# Subfossil Insects of the Kebezen Site (Altai Mountains): New Data on the Last Deglaciation Environment

**DOI:** 10.3390/insects16030321

**Published:** 2025-03-19

**Authors:** Anna A. Gurina, Natalia I. Agrikolyanskaya, Roman Yu. Dudko, Yuri E. Mikhailov, Alexander A. Prokin, Sergei V. Reshetnikov, Alexey S. Sazhnev, Alexey Yu. Solodovnikov, Evgenii V. Zinovyev, Andrei A. Legalov

**Affiliations:** 1Institute of Systematics and Ecology of Animals, Siberian Branch of Russian Academy of Sciences, Novosibirsk 630091, Novosibirskaya Oblast, Russia; auri.na@mail.ru (A.A.G.); natylnik@mail.ru (N.I.A.); rdudko@mail.ru (R.Y.D.); reshetnikov-art@yandex.ru (S.V.R.); 2Institute of Biology, Ecology and Natural Resources, Kemerovo State University, Kemerovo 650000, Kemerovo Oblast, Russia; 3Department of Ecology & Nature Management, Ural State Forest Engineering University, Yekaterinburg 620100, Sverdlovsk Oblast, Russia; yuemikhailov@gmail.com; 4Department of Earth and Space Sciences, Ural Federal University, Yekaterinburg 620002, Sverdlovsk Oblast, Russia; 5Papanin Institute for Biology of Inland Waters, Russian Academy of Sciences, Borok 152742, Yaroslavl Oblast, Russia; prokina@mail.ru (A.A.P.); sazh@list.ru (A.S.S.); 6Natural History Museum of Denmark at the University of Copenhagen, 2100 Copenhagen, Denmark; asolodovnikov@snm.ku.dk; 7Institute of Plant and Animal Ecology, Ural Branch of Russian Academy of Sciences, Yekaterinburg 620144, Sverdlovsk Oblast, Russia; zin62@mail.ru; 8Department of Ecology, Biochemistry and Biotechnology, Altai State University, Barnaul 656049, Altayskiy Kray, Russia; 9Department of Forestry and Landscape Construction, Tomsk State University, Tomsk 634050, Tomsk Oblast, Russia

**Keywords:** western Siberia, insects, palaeoenvironment, Late Pleistocene, last glacial maximum, MIS2, last deglaciation, megafloods, primary succession

## Abstract

This paper focuses on insect remains found at the Kebezen site in the Altai mountains, Russia. The results obtained from the samples supplement the information about the natural environment in Altai during the degradation of the last glaciation. The age of the studied sediments, from 20,100 to 19,300 cal yr BP, corresponds to the beginning of the degradation of the last glaciation. Beetles, represented by 105 species from 21 families, predominate in the site. The data obtained about these insects show that at this time, in contrast to the dry steppes of the West Siberian Plain, spruce forests and alpine meadows were developed in the north-eastern Altai region, and the climate was cold and humid. During sedimentation, rapid changes in the ecological composition of beetles were observed: the full-fledged forest complex was simplified and only meadow, and near-water species remained, and then species associated with shrubs, moss turfs, and trees consistently appeared. Such changes could be associated with a catastrophic event such as a megaflood.

## 1. Introduction

During the cold stages of the Pleistocene, in particular during the last glacial maximum, open steppe periglacial landscapes with xerophytic vegetation in dry and cold climates were widespread across Eurasia, and reticulated glaciation formed in the mountains [1,2]. At the same time, in some mountainous regions, in the transition zones of low mountains, mild and humid climate conditions were maintained during the Late Pleistocene. Those more favorable climatic conditions played an important role in the insect faunal changes during the Quaternary and the conservation of biodiversity in continental Asia. One such refugia has been reconstructed for the north-eastern Altai region, north of Lake Teletskoye [3]. For this region, palaeoenvironment reconstructions of the LGM period are very contradictory; inferred conditions vary from humid landscapes with boreal and even broadleaf forest to arid steppe and loess steppe environments [1,3,4]. Even more contradictions are associated with reconstructions of the last deglaciation of the Altai mountains. According to many researchers, it took place extremely rapidly, with catastrophic phenomena, and it was accompanied by multiple fluctuations in temperature and humidity [4,5,6]. According to a contrary point of view, it took place relatively slowly [2]. The difficulties in reconstructing the events of this time are due to both disagreements in the interpretation of the time and origin of traces of glaciations and the limited study of paleontological material.

Insects represent one of the most sensitive and reliable indicators of past environments due to their abundance and high degree of taxonomic and ecological diversity [7,8]. Together with other methods, palaeoentomology has long been successfully used in reconstructions of ancient landscapes and climate in the Quaternary period of western Siberia [9,10].

Previously, a unique Late Pleistocene “Otiorhynchus-type” fauna was discovered in the south of the West Siberian Plain, which has no modern analogs [11]. This fauna inhabited open tundra–steppe landscapes, with small forest patches preserved only under certain conditions [12]. Until recently, information on fossil insect complexes of the Altai–Sayan mountain system was nearly absent, with the exception of a few works [13,14] related to insects from the early and middle Holocene of the Altai–Xinjiang region. Investigations of the Late Pleistocene insect fauna of the north-eastern Altai region began with the study of deposits in the lower reaches of the Lebed River (a tributary of the Biya River) [15], where two insect assemblages were identified, corresponding to the Oldest and Older Dryas (~16.8 and ~13.8 cal ky BP). Based on them, Gurina et al. [15] reconstructed wet and cold conditions and plant communities with a predominance of alpine meadows and dark coniferous forests.

This paper explores another subfossil insect site that was found in a thickness of clayey sediments on the right tributary of the Biya River, 35 km upstream from the Lebed section, near the village of Kebezen, which, according to radiocarbon dating, corresponds to the beginning of deglaciation (Figure 1). The aim of this paper is to characterize the insect assemblages of the Altai foothills during the last glacial maximum in comparison with the modern and Late Quaternary insect fauna of the region in order to reconstruct the palaeoenvironmental conditions and to compare them with existing reconstructions of this time.

## 2. Regional Setting

### 2.1. Study Area

The locality of Kebezen (51.93600° N; 87.09665° E; 450 m above sea level) is situated on the left bank of the Turachak stream, 1.2 km from its mouth (where it flows into the Biya River) near the village of Kebezen (Figure 1). Sediment accumulation occurred in a small depression in the relief between Mount Chebor (779 m above sea level) and Mount Aktash (741 m above sea level) on the opposite bank of the Biya River, with this area being bordered by the small elevations of Kezekesh (518 m above sea level) and Kokaikha (674 m above sea level). At the mouth of the Turachak stream, the altitude does not exceed 395 m above sea level. The small altitudinal gradient from that mouth over a distance of 3.5 km (the length of the Turachak stream) and a small basin area (less than 5 km^2^) indicate the exceptionally local nature of sediment accumulation. Overall, the area where the Kebezen site is located is a highly eroded mid-altitude plateau (with heights ranging from 600 to 1000 m). The majority (about 70%) of its territory is covered by dark taiga, consisting of tall grassy aspen–fir and derived tall grassy birch–aspen forests with shrub thickets. In the Biya valley, intra-zonal pine forests predominate [16,17]. The area is characterized by a mild climate with abundant precipitation, even in the driest month (February), receiving up to 40 mm of rainfall. The highest amount of precipitation (187 mm) falls in July, and the average annual precipitation is 1344 mm. The average temperatures in the warmest month of July are +16.4 °C, while the lowest average temperatures in January are around –15.6 °C [18].

### 2.2. Geological Setting

The Quaternary history of the northern Altai region remains insufficiently studied, due in part to the weak and fragmentary exposures of Quaternary deposits, where only the upper parts of the sections are available for study. The vast majority of these sections date back to the last glaciation [4,6].

In the mountain basins of the Altai region (such as Chuiskaya and Kuraiskaya), large ice or moraine dammed lakes formed during the Pleistocene stadials [2,6,19]. When the climate warmed (or due to tectonic causes), the dams collapsed, and the lakes drained. At high rates of dam failure, catastrophic megafloods, also known as glacial lake outburst floods (GLOFs) occurred along the valleys of main rivers (for example, Chuya–Katun and Bashkaus–Chulyshman–Biya). Such flows carried large volumes of sediment of various fractions, from rocks to mud, and significantly influenced the characteristics of sediments in the valleys of the main rivers and their tributaries. One of the central controversies in understanding the ongoing processes is estimating the flow rate of megafloods. Particularly large disagreements are associated with the collapse of the largest Chuya–Kurai paleolake at the end of the last glaciation. Some researchers [6,19,20,21] believe that it drained instantly in geological terms, while others suggest a gradual release of water [2,22,23,24,25]. For example, according to the results of recent modelling work, 95% of the lake drained in about 33.8 h, the peak discharge of water reached 10.5 million m^3^/s, and the transit flow velocity reached 40 m/s [21]. According to other estimates, it is assumed that the lake drained over at least 200 years (estimated by the number of abrasion-accumulative terraces), with the maximum water flow during seasonal floods reaching 800–2120 m^3^/s. However, even this exceeds the current discharge of water at the mouth of the Chuya River by 20–50 times [25].

Late Pleistocene megafloods have also been reconstructed on the Biya River. They are associated with the rapid discharge of water from Lake Teletskoye, as well as from the paleobodies of water of the Chulyshman and Bashkaus basins [4,5,6]. The sixth fluvial terrace of the Biya (90–120 m above the current river level) in its upper reaches was formed from megaflood deposits during the failure of the terminal moraine of the valley glacier occupying the latitudinal section of Lake Teletskoye during the last glaciation [5,26]. The terminal moraine was located opposite the mouth of the Iogach River (Figure 1a,b), and the level of Lake Teletskoye was 210–220 m higher than today. The failure of the moraine is associated with a sharp warming of the climate and intensive melting of glaciers and is dated to the beginning of the last deglaciation—an interval of 20–16 ky BP (~24.0–19.3 cal ky BP) [5] or 17–15 ky BP (~20.5–18.3 cal ky BP) [4]. The probable cause of the initiation of the Teletskaya moraine breakthrough is the catastrophic collapse of the Tuzharskaya system of palaeolakes on the watershed of the Chulyshman River and Bashkaus River [4,6]. The discharge of water from Lake Teletskoye was catastrophic; the flow carried debris up to 8 tons and a volume of over 3 m^3^ (the maximum size of boulders on the sixth terrace of Biya), which made it possible to estimate the flow velocity at 7–7.5 m/s [5,26]. Part of the flow from the Biya valley crossed the low watershed (40 m) and passed along the Isha River valley [4].

As a result of the megafloods, the Biya valley was filled with a large amount of debris, which created conditions for damming its tributaries and formation of the so-called diluvial-dammed lakes in them [4]. Such lakes existed for different periods of time, but they eventually broke through the diluvial dam and caused new floods of significantly lesser power. Clayey sediments of diluvial-dammed lakes from different stages of the late glacial period have been found in most tributaries of the Biya and Isha (the rivers Pyzha, Sarakoksha, Turachak, Tuloy, Lebed, Uchurga, Kazha, Yugala, etc.) [4,5]. In some tributaries, lakes were formed several times; for example, Tuloy Lake was formed at least four times during the Dryas and was finally drained at the boundary of the Pleistocene and Holocene [4].

There are also other versions of the formation of fluvial terraces on the Biya River that are not associated with GLOFs. For example, Okishev [2] believes that the terminal glacial moraine at the headwaters of the Biya dates back to the middle Neopleistocene. He also correlates the formation of the highest fluvial terraces of the Upper Biya of kame genesis with this same period [2].

### 2.3. Description of the Sections

The lake sediment outcrop on the left bank of the Turachak stream, 1.1 km from the mouth, was first described by Baryshnikov in 1981 (Figure 1c and Figure 2b), but it was published somewhat later [5,27]. In the lower part of the 15 m natural outcrop, a 3.9 m thick layer of dark blue clay–silt deposits with interlayers of sandy loam, rich in plant remains, was revealed, and at the base, a 1.6 m layer of dark grey sandy loam, with remains of woody vegetation and seed flora, was identified by E.A. Ponomareva, which included the following: *Picea obovata* Ledb., *Pinus silvestris* L., *Larix* sp., Pinaceae gen. ind., *Betula nana* L., B. sp., *Alnus* sp., *Carex* ex gr. B, *Rumex* sp., Polypodiaceae gen. ind., *Viola* sp., *Aquilegia* sp., *Adoxa moshatellina* L., and Umbelliferae gen. ind. A radiocarbon date of 14.98 ± 0.07 ky BP or 18.23–18.28 cal ky BP (SOAN-576) was obtained for the layer.

We surveyed the Kebezen site on the Turachak stream in July 2019. Clearing of the exposure face was carried out at the point with the maximum natural exposure (height 17.15 m). The description (performed by E.V. Zinovyev) is given in Table 1 and Figure 2a and Figure 3a. Grey and blue clays predominate in the lower and middle parts of the section (layers 5–11), indicating an anaerobic sedimentation regime. These layers contain interspersed layers of plant detritus (mainly wood) of varying abundance, from which twelve samples were taken for entomological analysis. Six radiocarbon dates were obtained (Figure 2a). At the base of the section (layer 11), the alluvium of the Turachak stream (medium-grained sand) is interbedded with dark grey clays and includes a large amount of detritus. The overlying non-bedding light-grey clay (layer 10) is one meter thick and contains inclusions of woody remains (especially in the lower part) and a large number of mollusk shells. Layers 9 and 7 have rhythmic horizontal (or wavy) layering and consist of interlayers of light grey clays similar to layer 10 and darker clays (sometimes including small amounts of sandy loam and lenses of compacted plant detritus) that correspond to the floodplain facies. Above the section (layers 5 and 4), the layering ceases to be evidently rhythmic, and interlayers of iron-rich sediments appear. In the lower part of layer 5, cryoturbation was revealed, clearly expressed in an additional section made 10 m to the upstream of the main section (Figure 3b). In the upper part of the sections, lacustrine and lacustrine–floodplain deposits are covered by a layer of loam (layer 3).

## 3. Materials and Methods

### 3.1. Sampling

Samples for entomological analysis were taken from layers and interlayers rich in plant detritus using the standard methodology [28]. A total of 12 samples (S1–S12) were taken, with the samples numbered from bottom to top (Figure 2 and Figure 3, Table 2). Each sample was taken in several (1–7) replicates (subsamples). The volume and number of subsamples were determined based on the concentration of plant detritus and insect fragments in the deposits in order to obtain representative material. Samples S9 and S10, with a high content of plant detritus, were taken without prior washing in 1 L of sediment per subsample. The rest of the samples (with a volume of 5–10 L per subsample) were enriched in the field conditions using wet filtration through a sieve with a 0.3 mm mesh (Table 2).

### 3.2. Treatment

In the laboratory, the samples were washed with tap water and separated into fractions using wet sieving through screens with cell diameters of 2 mm, 0.6 mm, 0.3 mm, and 0.1 mm. The fractions were dried at room temperature. Insect fragments were picked out from them under a Carl Zeiss Stemi 2000 binocular microscope (Carl Zeiss Microscopy GmbH, Jena, Germany). Fragments of good quality (whole or half head capsules, elytra, pronotums, terminalia, etc.) were cleaned with a cleaning agent, and then more persistent contaminants were removed from the fragments using an ultrasonic bath (35 W, 10–20 s). The prepared fragments were glued to cardboard slides with water-soluble glue. Fragments that were “not suitable for identification” (exoskeleton scraps, abdomen sternites, legs, antennal segments, etc.) were placed in plastic containers for storage and were not considered in this study. The identification of the material was carried out by comparing the fragments with extant species. Collections from the following institutions were used for comparison: Institute of Systematics and Ecology of Animals Siberian Branch of RAS (Novosibirsk), Institute of Plant and Animal Ecology, Ural Branch of RAS (Ekaterinburg), Zoological Institute of RAS (St. Petersburg), Paleontological Institute of RAS (Moscow), Moscow Pedagogical State University, Papanin Institute for Biology of Inland Waters, Russian Academy of Sciences (Borok), and Natural History Museum of Denmark at the University of Copenhagen (Copenhagen, Denmark).

The *Nmin* (minimum number of individuals) indicator was used to assess the number of individuals of each species in the samples and subsamples [12,29]. Statistical analysis was conducted using the PAST 4.11 software. When comparing species composition, hierarchical clustering and standard correspondence analysis were used. Hierarchical clustering with the application of the Simpson pairwise similarity index and UPGMA linkage method was calculated according to the stratigraphic position of the samples and was validated with bootstrapping (10,000 iterations). The choice of the Simpson index, defined as the ratio of the number of common species to the number of species in a smaller sample, was based on its sensitivity to the actual substitution of species composition in samples and its ignorance of richness depletion, unlike most popular indices. The UPGMA method was also used, which favors clusters with high species richness, since depletion is likely to be due to taphonomic features rather than climatic or environmental fluctuations. Standard correspondence analysis was used for visual ordination of subsamples and species, as well as subsamples and ecological groups of Coleoptera. Untransformed data with abundance (*Nmin*) of species (or ecological groups) were used. Species represented in only one subsample, as well as a eurytopic group, were excluded from the analysis.

Photographs of insect fragments were taken with an AxioCam MRc5 camera (Zeiss) using a Carl Zeiss Stemi 2000 binocular microscope. Illustrations were prepared using Corel Draw 13.0 and Adobe Photoshop CS5.

All treated invertebrate remains are preserved in the collection of the Institute of Systematics and Ecology of Animals SB RAS (Novosibirsk).

### 3.3. Radiocarbon Dating

Radiocarbon dating was conducted using plant detritus (100 g each) from samples S2 (layer 11), S4 (layer 10), S8 (layer 10), S9 (layer 9), S10 (layer 9), and S11 (layer 7) at Herzen State Pedagogical University of Russia, St. Petersburg. The obtained radiocarbon dates were calibrated using the Calib Rev 8.1.0 software with the IntCal20 curve, within a range of ±2σ (Table 2). Additionally, a previously published date from the base of the Turachak section [27] was used in the study.

Based on the dates obtained in the Bchron 4.7.6.9000 package [30] in the R 4.4.2 software (www.r-project.org, access date 10 January 2025), an age–depth model was constructed with default settings as follows: an IntCal20 curve for calibration, 10,000 iterations, a slightly modified Markov chain with the Monte Carlo fitting algorithm. The resulting curve was interpreted as a linear interpolation.

## 4. Results

### 4.1. Entomological Data

A total of 1564 insect fragments were processed from the Kebezen site. In addition to insects, 26 carapaces of spiders of the family Linyphiidae were picked out, and singletons of Oribatida mites and numerous gastropod shells were recorded. Although Branchiopoda (Cladocera and Ostracoda) were also targeted when sorting the samples (Table 3), they were not found. Among the insects, 1405 fragments belonged to Coleoptera, 10 to Hemiptera, 44 to Hymenoptera, and 72 to Diptera (puparia). Mostly, representatives of Coleoptera were used in the analysis here. The studied insect assemblages were characterized by a high degree of fragment association, meaning that fragments belonging to the same individual were found in various samples, indicating weak sediment movement in the water body where the insect remains accumulated.

The beetle fragments were attributed to at least 772 individuals from 21 families, belonging to no fewer than 105 species, of which 70 were identified to the species level (Table A1). The most numerous families were Carabidae (28 species, *Nmin* = 290), Staphylinidae (24 species, *Nmin* = 160), Chrysomelidae (8 species, *Nmin* = 101), and Scolytidae (5 species, *Nmin* = 53). The most numerous species in terms of individuals was the ground beetle *Bembidion difficile*, *Nmin* = 46. Several other species from the families Carabidae, Heteroceridae, Scolytidae, and Hydraenidae were also numerous, such as *Trechus* cf. *toroticus*, *Nmin* = 30; *Augyles intermedius*, *Nmin* = 26; *Carphoborus teplouchovi*, *Nmin* = 26; *Pterostichus* cf. *triseriatus*, *Nmin* = 22; and *Ochtebius kaninensis*, *Nmin* = 22. The Simpson diversity index (1–*D*) was 0.9743. The relatively high species diversity was also reflected in the Shannon index values (*H* = 4.064) (Table 4).

A comparison of the species similarity of the samples and subsamples is presented in Figure 4. It shows the distinctiveness of the samples belonging to layer 10 (S4, S5, S6, S7, S8) and a high similarity between samples from layers 7, 9, and 11. The distinctiveness of S12 taken from layer 5 may be explained by the small sample size (20 fragments, *Nmin* = 5).

Through cluster analysis and taking into account the stratigraphy, five main clusters were distinguished, corresponding to layers 11, 10, 9, 7, and 5 (Figure 5). As an exception, sample S4, taken from the lower part of layer 10, based on the species composition of Coleoptera on the dendrogram, fell into one cluster with samples from layer 11. However, the average similarity coefficient of species between sample S4 and S1–S3 was 55.8%, and that between samples S4 and S5–S8 was 54.5%, i.e., sample S4 in terms of beetle composition occupied an intermediate position between the complexes of layers 11 and 10.

### 4.2. Radiocarbon Dates

Calibration of the five radiocarbon dates obtained from the layers 11–9 yielded virtually identical results, nearly 600-year ranges (20.078–19.503 cal ky BP), and a somewhat younger age for layer 7 (19.59–19.12 cal ky BP), although this range also partially overlapped with the previous ones (Table 2, Figure 6).

The age–depth model constructed using these dates showed an average sedimentation rate of about 1 cm per year, although the 2σ accuracy allows for an interpretation of the deposition time of a 2.5 m thickness from instantaneous to 700 years (Figure 6).

## 5. Discussion

### 5.1. Taxonomical Composition and Diversity

The complex of beetles from the Kebzen site is characterized by a significant number of identified species (105 species, 70 determined to the species group), which are relatively evenly distributed over a sediment thickness of more than three meters (Table 4). This allows for a sufficiently representative comparison of this insect assemblage with other fossil and modern faunas, as well as an analysis of the dynamics of the fauna that existed during the formation of the deposits.

The large number of identified species and families of Coleoptera already allows us to conclude that the environmental conditions were quite diverse and never extremely harsh despite the fact that the formation of the deposits occurred in the cold period near high mountains. It is worth noting the occurrence of the families Cantharidae, Dermestidae, and Laemophloeidae, which are rarely found in Quaternary deposits, especially Ciidae and Pythidae, which are here recorded in the Pleistocene of western Siberia for the first time.

Even more indicative are the relatively high values of the Shannon and Simpson diversity indices (Table 4), which remained high in most samples (except for S5, S6, and S12). At the same time, their values were somewhat lower than those for the beetle assemblage from the nearest known locality, Lebed (where *H* = 4.72, 1–*D* = 0.988) [15]. The greater diversity of this assemblage may be due to its geographic location closer to the periphery of the mountain system. The moderate degree of dominance, not exceeding 22% in individual samples and 5% for the Kebzen insect assemblages as a whole, and the presence of several dominant species also confirm the relatively mild natural conditions during sediment accumulation.

### 5.2. Faunal Comparisons

#### 5.2.1. Comparison with Late *Pleistocene fauna* of Western Siberia

Almost half (31 out of 70) of the determined species from the Kebzen site have not been previously recorded in the Pleistocene deposits of western Siberia (marked with an asterisk in Table A1 and Figure 7, Figure 8 and Figure 9), despite the relatively well-studied fauna of Late Pleistocene beetles in the West Siberian Plain (over 400 species) [10,12,15,28]. The majority of new species in the Pleistocene were found in the families Carabidae (10 species), Chrysomelidae (4), and Curculionidae (3). Two species were added to the list of Leiodidae and Staphylinidae, and one each was added to the list of Helophoridae, Hydraenidae, Scarabaeidae, Heteroceridae, Elateridae, Cantharidae, Dermestidae, Brentidae, Scolytidae, Ciidae, and Pythidae. The last two families have never been found in the Pleistocene of the region. Unusual for Western Siberia is the representation of the main families, namely the low proportion of Curculionidae (10 species, 11%), which usually dominate in insect assemblages in the south of western Siberia. It is also unusual to have a high abundance of fragments of Chrysomelidae, Scolytidae, Hydraenidae, and Heteroceridae.

Such a high degree of novelty is probably determined by the geographical location of the site in the foothills of the north-eastern Altai region. From adjacent territories, only two insect sites are known: the partially processed site Novaya Surtaika (MIS 2) [31] and Lebed (Oldest and Older Dryas) [15]. The Kebzen beetle complex shows a high similarity with the latter (36%). Despite the different ages, 24 common species from 10 families are noted with the Lebed beetle fauna: Carabidae (*Nebria gyllenhali*, *Carabus* cf. *aeruginosus*, *Elaphrus angusticollis*, *Bembidion difficile*, *Pterostichus brevicornis*, *P. maurusiacus*, *P. ehnbergi*, *P. subaeneus*/*P. monticoloides*, *Agonum alpinum*), Helophoridae (*Helophorus sibiricus*), Hydraenidae (*Ochthebius kaninensis*), Scarabaeidae (*Psammoporus matalini*), Byrrhidae (*Byrrhus* cf. *mordkovitshi*, *Simplocaria semistriata*), Elateridae (*Hypnoidus rivularius*-group), Cantharidae (*Podabrus alpinus*/*P. annulatus*), Brentidae (*Hemitrichapion tschernovi*, *Eutrichapion rhomboidale*), Curculionidae (*Tournotaris bimaculata*, *Phyllobius pomaceus*, *Trichalophus maklini*, *Otiorhynchus grandineus*), and Scolytidae (*Phloeotribus spinulosus*, *Polygraphus polygraphus*). Approximately the same similarity (36%, nine common species) is observed between the Kebzen and Novaya Surtaika insect faunas.

Thirteen insect sites of MIS 2 age are known from the territory of the West Siberian Plain, located from 68° to 51° N: Ngoyun, Agansky Uval-290/2, Agansky Uval-1082/2, Zeleny Ostrov-1, Bolshaya Gorka, Lokosovo, Malkovo, Dubrovino, Bunkovo, Suzun-1, Suzun-2, Smolenskoye, and Kizikha-1 [10,12,15,28]. Despite the similar age, the Kebzen site shows only a small similarity to these sites (no more than 15%), and the number of common species does not exceed 10.

Thus, the insect assemblages of the Kebezen, Lebed, and Novaya Surtaika sites demonstrate significant differences between the mountain ecosystems of the north-eastern Altai region and the entomofauna of the West Siberian Plain of the Late Pleictocene.

#### 5.2.2. Comparison with Modern Beetle Fauna

The species composition of the Kebzen site corresponds well with the modern fauna of the north-eastern Altai region and includes several endemic species. For example, *Carabus obovatus*, *Trechus toroticus*, *T. lomakini*, *Bembidion sajanum*, *Pterostichus triseriatus*, *P. monticoloides*, *P. subaeneus* (Carabidae), and *Byrrhus mordkovitshi* (Byrrhidae) are considered endemics, while *Pterostichus maurusiacus*, *P. ehnbergi* (Carabidae), and *Aegialia matalini* (Scarabaeidae) are considered subendemics of the Altai–Sayan mountain system.

Only a few species not known in the modern Altai fauna are noted in the Kebzen insect assemblages. For example, *Sternoplatys fulvipes* and *Denticollis acuticollis* currently inhabit areas significantly further east of the Altai region, in Transbaikalia and/or the Russian Far East. The species *Sulcacis nitidus* is widely distributed in the European part of Russia and is also recorded in the Russian Far East and Japan. *Ochthebius kaninensis* is common in north-eastern Europe and has been sporadically recorded in eastern Siberia. Additionally, the Palearctic species *Arpedium brachypterum* and *Olophrum fuscum*, not known from the Altai territory, are present in the beetle assemblages. Therefore, the Kebzen insect assemblage contains species whose current range is either limited to the Russian Far East or has wide Palearctic ranges that do not extend to the Altai region. The modern distribution of some of these species is insufficiently studied, and it is possible that they will be found in the Altai region. However, the presence of these species in the studied assemblage may indicate their wider distribution during the Late Pleistocene.

### 5.3. Ecological Analysis

#### 5.3.1. Altitudinal Distribution

Most of the species from the Kebzen site are widely distributed in the region along an altitudinal gradient. Due to the currently insufficient level of knowledge of the modern distributions of Altai beetles, it is not possible to conduct a detailed analysis. Several high-mountain species that are currently absent from the area of the Kebezen site were found in the deposits. These are species of alpine meadows (*Trechus lomakini*, *T. toroticus*, *Coccinella nivicola*), mountain tundra (*Diacheila polita*, *Bembidion dauricum*, *Pterostichus brevicornis*, *P. burjaticus*, *P. fulvescens*, *P. subaeneus*, *Byrrhus mordkovitshi*, *Hemitrichapion tschernovi*), or the shores of high-mountain lakes (*Bembidion bipunctatum*). They are usually found above 1500 m a.s.l. in the region, and some have been recorded at minimum altitudes of about 1000 m a.s.l. [32,33,34]. The presence of such species in the sediments indicates significantly colder conditions during the sedimentation period than now. High-mountain species are fairly evenly represented in all samples and make up from 12 to 40% (on average 24.5%) of the identified species. As an exception, in sample S11, only two species of the high–mid-mountain complex were recorded, namely *P. fulvescens* and *Notiophilus fasciatus* (Figure 10).

#### 5.3.2. Trophic Groups

The Coleoptera assemblage from the Kebzen site includes beetles with diverse trophic requirements. Predatory beetles are the most numerous group, represented by more than 50 species, mainly from the families Carabidae and Staphylinidae. Among phytophagous beetles, at least 25 species develop on higher plants. Algal feeders, bryophagous, mycetophagous, and saprophagous beetles are also sporadically represented.

Predatory Coleoptera were quantitatively (*Nmin*) predominant in the majority of the samples and constituted ca. 60%. In sample S5 and especially S4 and S12, their share increased to 71–85%. In contrast to this, their share decreased to ~50% in samples S8 and S10 (Figure 11). A high proportion of predatory beetles is characteristic of cold and humid conditions, as well as for the pioneer communities living in highly disturbed habitats [35,36,37].

For reconstructions of conditions, phytophagous beetles, especially those specific to certain plants, are of significant interest. They are represented by the following four subgroups:Species developing on conifers, predominantly *Picea*, less frequently *Pinus*: *Pissodes gyllenhali*, *Xylechinus pilosus*, *Carphoborus teplouchovi*, *Polygraphus*, and *P. subopacus*. No species obligate to other species of coniferous and deciduous trees were identified.Species associated with near-water herbaceous vegetation (*Phaedon armoraciae*, *Tournotaris bimaculata*, *Notaris acridulus*, *Ceutorhynchus cochleariae*) and shrubby vegetation (*Lochmaea caprea*, *Phratora* sp., *Gonioctena* spp., *Dorytomus taeniatus*). These species indicate the presence of willow (possibly *Salix aurita*, *S. dasyclados*, *S. myrsinifolia*, *S. caprea*), Brassicaceae (*Cardamine* sp.), Ranunculaceae (*Caltha* sp.), Plantaginaceae (*Veronica* sp.), and Poaceae (*Glyceria* sp., *Phalaris arundinacea* sp.) in the area.Species associated with meadow vegetation (*Sternoplatys fulvipes*, *Hemitrichapion tschernovi*, *Eutrichapion rhomboidale*, *Sitona lineellus*, *Boreohypera diversipunctata*, *Phyllobius pomaceus*, *Trichalophus maklini*, and *Otiorhynchus grandineus*). These species indicate the presence of Fabaceae (*Lathyrus gmelini*, *Oxytropis* sp., *Vicia* sp., *Trifolium* sp., Medicago sp.), Caryophyllaceae (*Cerastium* sp., *Stellaria* sp., *Silene* sp.), Grossulariaceae (*Ribes* sp.), Urticaceae (*Urtica* sp.), and Rosaceae (*Filipendula* sp.) in the area.Species trophically associated with algae or moist detritus, such as *Helophorus praenanus*, *H. sibiricus*, *Ochtebius flavipes*, and *O. kaninensis*.

Reconstructed herbaceous and shrubby plant taxa based on trophic specificity of Coleoptera are characteristic elements of modern dark coniferous forests and alpine meadows. However, in the modern composition of trees near Lake Teletskoye, *Abies*, *Pinus*, and *Betula* dominate, while *Picea* is significantly inferior in number [16,17].

#### 5.3.3. Biotopic Groups

The distribution of species into ecological groups was primarily based on their preferred habitats in the Altai region. Twelve biotopic groups were identified, which were classified into four types. The first type included 23 species of Coleoptera associated with water. They were divided into four groups based on their degree of association with water, ranging from strictly aquatic to preferring moist habitats (Table 5). The second type consisted of species found in open landscapes. In the Kebezen area, only seven species were included in this type, which were further divided into three groups (dry meadows, alpine meadows, and tundra). The third type included species living in forest communities or obligately associated with woody vegetation, totaling 18 species. The fourth type comprised species characteristic of both open and forest landscapes (21 species). In addition to eurytopic species, this type also included species typical of the middle and high mountain ranges of the Altai region, penetrating into the forest belt up to the foothills to varying degrees. These species were divided into two groups: boreo-alpine, inhabiting alpine meadows and the herbaceous layer of the forest, and forest–tundra, inhabiting mountain and zonal tundras and the forest zone, preferring river floodplains and/or the coldest habitats in the latter.

The ratios of ecological groups of insects in sediments correlate well with changes in environmental conditions [38]. Figure 12 shows the distribution of beetle abundance of the different ecological groups across the samples. In the samples of layer 10, the water-edge group (I-3) and boreo-alpine group (IV-1) are well represented. In contrast, forest species are practically absent here (only sporadically present in sample S5), both xylomycetophagous species (III-2) and those not directly associated with trees (III-1). These same patterns are reflected and statistically supported by correspondence analysis considering the distribution of ecological groups across subsamples (Figure 13). There were no differences in the ecological composition of beetles during the formation of layers 11, 9, and 7, as the dispersion among subsamples is comparable to that among samples and layers. The dynamics of the ecological composition of beetles are discussed in the next chapter, in conjunction with the dynamics of other parameters.

According to the ecological composition of Coleoptera of the Kebezen site, it is similar to the Lebed site, with a characteristic predominance of taiga and alpine species [15]. However, unlike the latter, there is a greater proportion of tundra species in Kebzen and a complete absence of steppe elements.

The high proportion of forest species, low proportion of open landscape species, and absence of steppe and halobiont species can be considered as characteristic features of Late Pleistocene insect assemblages in the north-eastern Altai region, distinguishing them from those of the West Siberian Plain and those of north-eastern Siberia. These differences are due, first of all, to the significantly higher humidity of the north-eastern Altai region against the background of the general aridity of the continental regions of Eurasia [1,3].

### 5.4. Conditions for the Formation of the Kebezen Section

The studied thickness of the lake sediments is highly unusual. First of all, attention is drawn to a series of five almost coinciding radiocarbon dates (from 16.407 ± 0.1 ky BP to 16.353 ± 0.1 BP) from an almost three-meter layer of clayey sediments in the lower part of the section (layers 11–9) (Figure 2). The age–depth model constructed using these dates shows an unusually high average sedimentation rate of about 1 cm per year (Figure 6). The common rate of accumulation of lacustrine clays, even in modern periglacial areas, is rarely more than 2 mm per year, and more often, a meter-thick layer accumulates over ~1000 years or more [39,40,41]. The pronounced dynamics of the species and ecological composition of Coleoptera extracted from these layers also indicate changes in the environment during the sedimentation period and reinforces the contradictions with the dating.

The reliability of the radiocarbon dates is confirmed by the following arguments: (1) the good convergence of these dates obtained from wood remains; (2) their proximity to previously obtained dates from the deposits of dammed lakes of some other tributaries of the Biya, i.e., 16.12 ± 0.08 ky BP (19.18–19.61 cal ky BP) on the Pyzha River, 16.19 ± 0.09 ky BP (19.28–19,84 cal ky BP) on the Uchurga River, and diluvium in the upper reaches of the Isha River (between 20.5 and 18.3 cal ky BP) [4,27]; and (3) the association of these dates with one event, the failure of the terminal moraine of Lake Teletskoye [4,5].

This contradiction can be explained by the peculiarities of the formation of the paleolake on the Turachak stream, associated with the catastrophic megaflood of the Biya River. Based on this reconstruction of events, the catastrophic flows formed a dam of large-boulder material at the mouth of the Turachak stream, and the mudflow fraction flooded the valleys of the Biya tributaries at significantly higher altitudes [5]. Spring waters washed away mud (mudflow deposits from the megaflood) from the slopes of the Turachak valley, which contributed to the high turbidity of the water in the emerging lake and the high rate of sedimentation. The high rate of sedimentation in the Yugalinskoye palaeolake (a tributary of the Isha River) and other lakes that formed during the main phase of glacier degradation was noted by Rusanov [4]. In addition, at the base of the section of the paleolake Pyzha deposits, which had a similar genesis, Baryshnikov [5] recorded varved clays with a varve thickness of up to 2–3 cm. Butvilovsky also believed that during the period of degradation of the Altai mountain glaciers, the rate of sedimentation was an order of magnitude higher than the normal level, and the thickness of the annual cycles averaged 10–30 cm per year [6]. It should also be noted that some older dates, compared to the actual age of the deposits, may be associated with the mass death of trees as a result of the megaflood and flooding of the resulting dammed lake.

Taking these factors into account, when interpreting the changes in the composition of the Coleoptera assemblages, one should keep in mind not only the climatic changes characteristic of deglaciation but also the consequences of the megaflood, such as significant degradation of soil and vegetation, as well as the presence/absence of a connection with the Biya floodplain.

#### 5.4.1. The Lower Strata of Lacustrine and Floodplain Sediments (Layers 11, 10)

Layer 11 forms the base of the section; it is an alternation of medium-grained alluvial sands and clays (samples S1, S2, S3). There is a high probability that this layer includes beetles transported by the megaflood from the Biya Valley. All ecological groups of beetles identified at the Kebezen site are present here (Figure 12). A high proportion of forest species, both xylobionts (spruce bark beetles, *Megatoma* cf. *undata*, *Leptophloeus alternans*) and those not associated with wood (*Notiophulus fasciatus*, *Pterostichus maurusiacus*, *Sternoplatys* cf. *fulvipes*), is characteristic, and there is also a noticeable inclusion of cryophilic species associated with moss turfs (*Diacheila polita*, *Pterostichus* (*Cryobius*) spp.). Contrary to this, the group of water-edge species is relatively poorly represented. It is interesting to note the relatively good representation of coprophages of the genus *Aphodius*, which are almost absent in other layers. This may indicate a long-term reduction in ungulate grazing after the megaflood.

Layer 10, non-bedded clay (samples S4–S8), corresponds to deposits of a stagnant (or rather weakly flowing) lake. As shown above, the insect assemblages of this layer contrast sharply with the rest, and the interpretation of these differences is of greatest interest.

In terms of the trophic composition, sample S4 stands out sharply due to the dominance of predatory beetles (mainly Staphylinidae and the ground beetles *Bembidion difficile* and *Agonum alpinum*) (Figure 11). A significant dominance of predators with the inclusion of detritivores is generally characteristic of the initial stages of primary succession and is known as “the predator’s first paradox” [36,37,42,43].

The biotopic confinement of the beetles of layer 10 (Figure 12 and Figure 13) is even more unique. It is best characterized by the rarity of forest species, present as singletons. This is despite the presence of woody remains throughout the entire thickness of the layer. It is likely that the plant remains were either redeposited from the megaflood mudflow deposits or were deposited here as a consequence of wood decomposition from the flooded forest. The lower part of the layer (samples S4–S6) also lacks species associated with moss turf. They appear only in samples S7 and S8 (Carabidae: *Pterostichus brevicornis*, *P.* cf. *burjaticus* and Byrrhidae: *Byrrhus* cf. *mordkovitshi*).

This layer, as a whole, is characterized only by two ecological groups: meadow species (typical of alpine meadows and grassy vegetation of forests) and the water-edge complex. Of the former, *Trechus* spp. and *Agonum alpinum* are particularly abundant; the former is evenly represented in the thickness of the layer, whilst the latter is more abundant in the lower part. The water-edge and floodplain complexes are especially numerous in the upper part of the layer and are represented by species characteristic of soft soils (clays, loams), such as *Augyles intermedius* (Heteroceridae), as well as those associated with water-edge grasses and willows, for example, *Phaedon* cf. *armoraciae* and *Phratora* sp. (Chrysomelidae).

Water beetles are represented by only two species of the genus *Ochtebius*. They live in shallow water, most often in temporary puddles or small streams, and prefer soft soil. Apparently, the diluvial-dammed lake itself remained oligotrophic throughout the entire period of sedimentation and was uninhabitable for invertebrates. This is confirmed by the absence in the sediments of not only lake species of insects but also daphnia and ostracods. The great depth combined with the small lake size, low temperatures, high turbidity, and too a short period of existence of the lake could be the reasons for the absence of invertebrates.

#### 5.4.2. The Middle Strata (Layers 9–7)

The appearance of rhythmic horizontal bedding characteristic of the floodplain facies indicates the connection of the diluvial-dammed lake with the Biya River during the flood. Such floods could have only been caused by large ice jams. In the Biya valley, traces of ice jams were destroyed by megafloods but they were probably very strong during stadials. Thus, in the Charysh valley in the north-western Altai region, during the maximum of the last glaciation, the water level rose by 20–40 m [4].

Probably, after the first flooding, the height of the dam decreased, and flooding became quite regular. Fluctuations in the lake levels undoubtedly affected coastal biota. In addition, the resulting flood-jam bodies of water could have existed for a long time (up to several months), destroying the riparian vegetation. Finally, the flood waters brought additional organic matter and insect remains. In other words, the area of flushing biogenic material into the sediments increased. All this influenced the composition of insects in the sediments.

The insect assemblages in samples S9, S10, and S11 were similar to each other and to the assemblages of the basal layer of the section (samples S1 and S2) (Figure 4). Taiga species reappeared, with xylobionts associated with spruce being particularly numerous. There were also numerous cryophilic species living in moss turfs (*Notiophilus fasciatus*, *Pterostichus* (*Cryobius*) spp., *Byrrhus* cf. *mordkovitshi*). In contrast, many meadow and, especially, water-edge species (*Trechus* cf. *toroticus*, *Agonum alpinum*, *Augyles intermedius*, *Phratora* sp.) disappeared or became rare. The representation of shallow-water aquatic beetles (genus *Ochtebius*) was uneven. They were quite numerous in samples S10 and S11 but were absent from sample S9.

#### 5.4.3. The Upper Strata (Layers 5 and 4)

The rhythmic layering of the clayey deposits of the lacustrine floodplain deposits is complicated by the fine bedding of dark inter-layers of lacustrine facies, inter-layers of fine sand, spots, and inter-layers of ferrugination, as well as cryoturbation in the lower part of layer 5. From a single sample, a lens of plant detritus from the upper part of layer 5 and only a limited number of beetle fragments were obtained. The presence of two cryophilic species (*Pterostichus* cf. *fulvescens* and *Byrrhus* cf. *mordkovitshi*) in such a small sample generally confirmed the conditions of a humid and cold climate, but it was insufficient for a more detailed analysis.

### 5.5. Environment Reconstruction and Age Correlation

The events reflected in the studied sediment layer probably followed immediately after the failure of the terminal moraine of the glacier of the last glaciation maximum at the source of the Biya and the catastrophic collapse of Lake Teletskoye. In turn, the breakthrough of the moraine is associated with the main phase of glaciation degradation—the warm period, when, according to Rusanov [4], summer temperatures in northern Altai were 4 °C higher than today.

However, the entomological data obtained indicate cold and wet conditions throughout the sedimentation period studied. Not a single thermophilic or xerophilic species of insect was found in these deposits, despite the fact that the section is located on the right bank of the Biya River, which has the warmest southwestern exposure. The obtained reconstructions are in good agreement with data on the macroremains of plants, determined by E.A. Ponomareva, from similar-age deposits of diluvial-dammed lakes on the Pyzha River [5] and Uchurga River [4]. Considering the radiocarbon age of the deposits (20.05–19.30 cal ky BP), they can be attributed to the end of the first phase of deglaciation and the beginning of the Oldest Dryas. The uppermost undated sample S12 studied appears to be from the Oldest Dryas.

The dynamics of the changes in the composition of the insects described above reflects changes in environmental conditions. The most complete spectrum of ecological groups of the basal layer (samples S1–S3) is replaced by a complex with a predominance of predatory beetles, typical of alpine meadows (sample S4). Then, in samples S5–S8, the proportion of water-edge beetles increases, including phytophagous species associated with willows and species inhabiting moss turfs. In the following layers (samples S9–S11), taiga species appear again (and dominate), while the proportion of water-edge and meadow species decreases, and the proportion of species associated with moss turfs increases. This course of changes corresponds well with primary successions associated with severe damage to the soil and vegetation (such as glacier retreat, large mudflows and avalanches, clearings of forests, etc.). It is likely that the megaflood and subsequent formation of the dammed lake could have caused similar changes.

For example, in the Tsey Gorge of the Central Caucasus, the succession of invertebrates associated with the retreat of the glacier [44] revealed a similar pattern. Predatory beetles from the families Carabidae and Staphylinidae (mainly the genera *Bembidion* and *Geodromicus*) appear immediately after the soil surface is freed from ice and become dominant during the first four years. On a seven-year-old plot, meadow species are added to the dominants, and a full-fledged complex, including forest species and moss turf species, is formed within 30–60 years. A similar pattern of changes in plant communities was observed after large mudflows. Thus, in the forest zone of the north Caucasus, on the large debris fan, the formation of diverse grass vegetation was noted after 2–5 years, and after 6–9 years, shrubs and trees appeared [45]. A large mudflow that passed through the central Altai region in the Aktru River valley in 1984 led to the destruction of the lower tiers of vegetation. Over the course of 32 years, a mixed-herb–green moss vegetation has formed there, similar in composition to the neighboring undisturbed forest [46]. Somewhat longer periods (45–60 years) are required for the restoration of stable phytocoenoses after the mudflows of the north Caucasus [45]. It should be noted that the impact of a megaflood on the soil and vegetation must be significant and large-scale; therefore, the restoration of the primary environmental conditions may require more time than in the examples given above, which is comparable to the time of deposition of the corresponding layers of the Kebezen site (Figure 6). Thus, it can be assumed that the changes in the insect assemblages of the studied strata reflect significant degradation of spruce forests and soil and vegetation in the Biya valley and its tributaries, associated with the megaflood and the formation of dammed lakes, as well as their gradual restoration through the stages of meadows and shrubs.

It should be noted that other interpretations of the observed changes in insect complexes are also possible. The increase in the proportion of tundra species in the upper part of the section may be due to the cooling of the climate during the transition to the Oldest Dryas. However, this point of view is contradicted by the absence of obvious dynamics in the representation of high-mountain species, which make up about 25% of all samples of the strata (Figure 10).

The presence of forest species in the lower and upper layers and their absence in layer 10 can be interpreted as the development of spruce forests predominantly in the Biya valley, while open-meadow landscapes predominate on the slopes. Therefore, forest species are recorded only in the layers involving Biya sediments.

## 6. Conclusions

The composition of the beetle fauna of the Kebezen site is unique and differs from both the modern fauna of the region and the known Late Pleistocene insect assemblages of western Siberia. One of its distinctive features is the composition of beetle families, with Carabidae, Staphylinidae, Chrysomelidae, and Scolytidae being the most numerous in this case. Such a composition most closely corresponds to the modern fauna of the middle elevation of the mountains of north-eastern Altai and includes endemic species of the Altai–Sayan mountains. Several species from the deposits are absent in the modern fauna of the region. The species composition of the Coleoptera of the Kebezen location is similar (36%) to the insect assemblage of Lebed and Novaya Surtaika from the Late Pleistocene of northern Altai and significantly differs from the steppe periglacial faunas of West Siberian Plain of that time.

It includes taiga species, especially various spruce xylobionts, as well as species characteristic of alpine meadows and mountain tundra. The intrazonal complex of the Kebezen assemblage is represented by riparian species associated with soft ground and near-water vegetation, i.e., marshland and shallow water species. There are no steppe or thermophilous elements in the studied assemblage. Based on entomological data, the conditions of a humid, cold climate are reconstructed, which contrast sharply with the arid climate of the Late Pleistocene of continental Asia as a whole.

The studied sedimentary layer of the paleolake on the Turachak stream, including entomological material, was formed over a short period of time (in the order of hundreds of years). Considering the radiocarbon dates of the studied sediments at 20.05–19.30 cal ky BP, previously known dates of sediments of a similar origin dammed lakes on the tributaries of the Biya, and the reconstructed conditions, the described events can be attributed to the end of the first phase of deglaciation and the beginning of the Oldest Dryas.

Despite the short period of formation of the deposits, they show clear dynamics of the composition of beetles. The most complete spectrum of ecological groups of the basal layer is replaced by a complex with a predominance of meadow species, and then the proportion of water-edge species increases, including phytophagous species associated with willows, and the appearance of species inhabiting moss turfs. In the following layers, the proportion of moss turf species increases, the taiga species appear (and dominate), while the proportion of the riparian and meadow species decreases. This series of changes can be interpreted as a succession initiated by the megaflood and the subsequent formation of a dammed lake. The impact of the coming cold weather cannot be ruled out.

## Figures and Tables

**Figure 1 insects-16-00321-f001:**
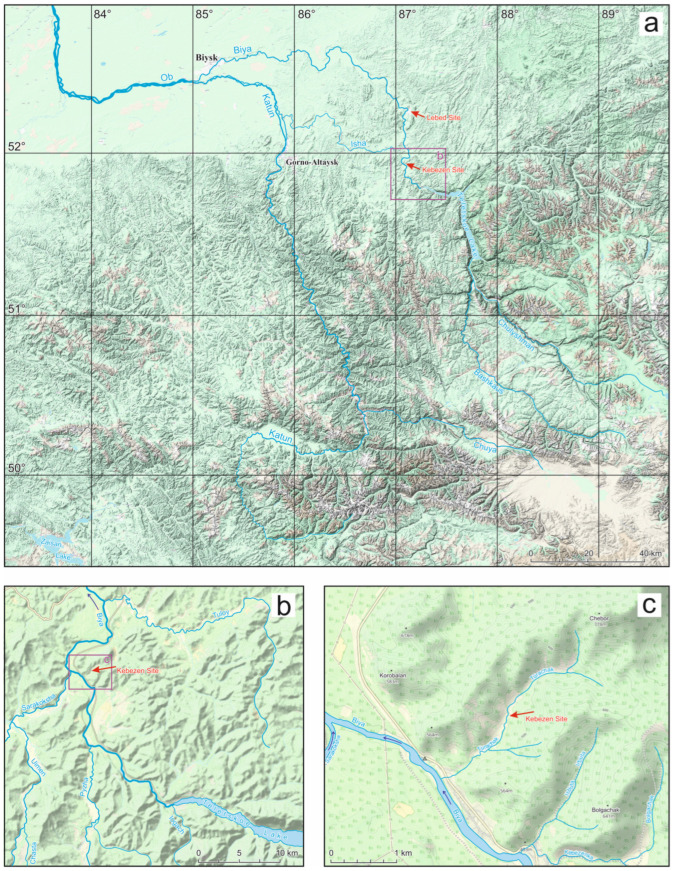
Location of the Kebezen and Lebed sites in the maps of Altai of different scales: (**a**)—small scale, (**b**)—medium scale, (**c**)—large scale.

**Figure 2 insects-16-00321-f002:**
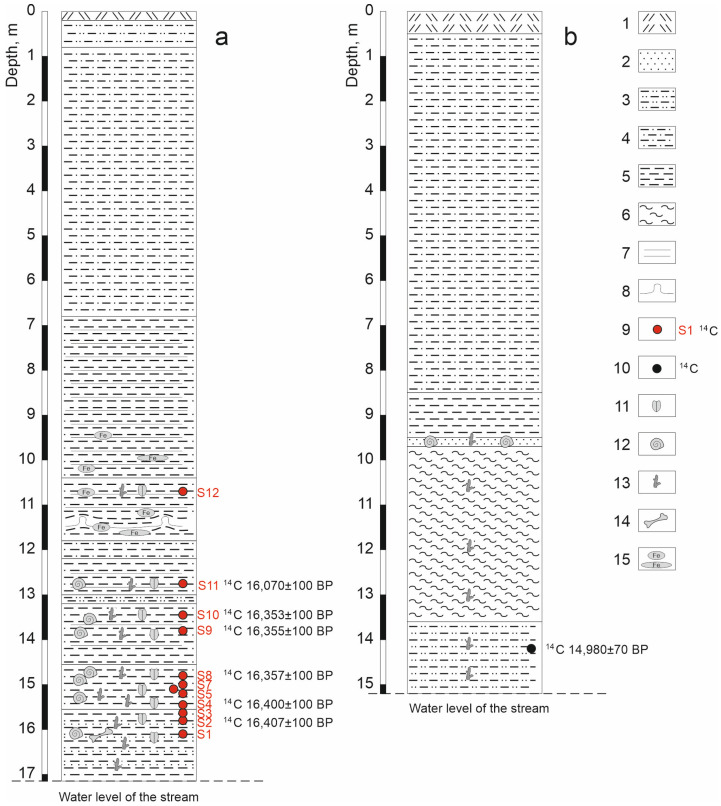
Stratigraphy of the Kebezen section (**a**) and a previously described [27] section on the Turachak stream (**b**). 1—Modern soil, 2—fine-grained sand, 3—sandy loam, 4—loam, 5—clay, 6—silty clay, 7—horizontal bedding, 8—cryoturbation, 9—samples and original radiocarbon dates, 10—radiocarbon date, according to [27], 11—insect remains, 12—mollusk shells, 13—plant detritus, 14—bones of small mammals, 15—spots and interlayers of ferrugination.

**Figure 3 insects-16-00321-f003:**
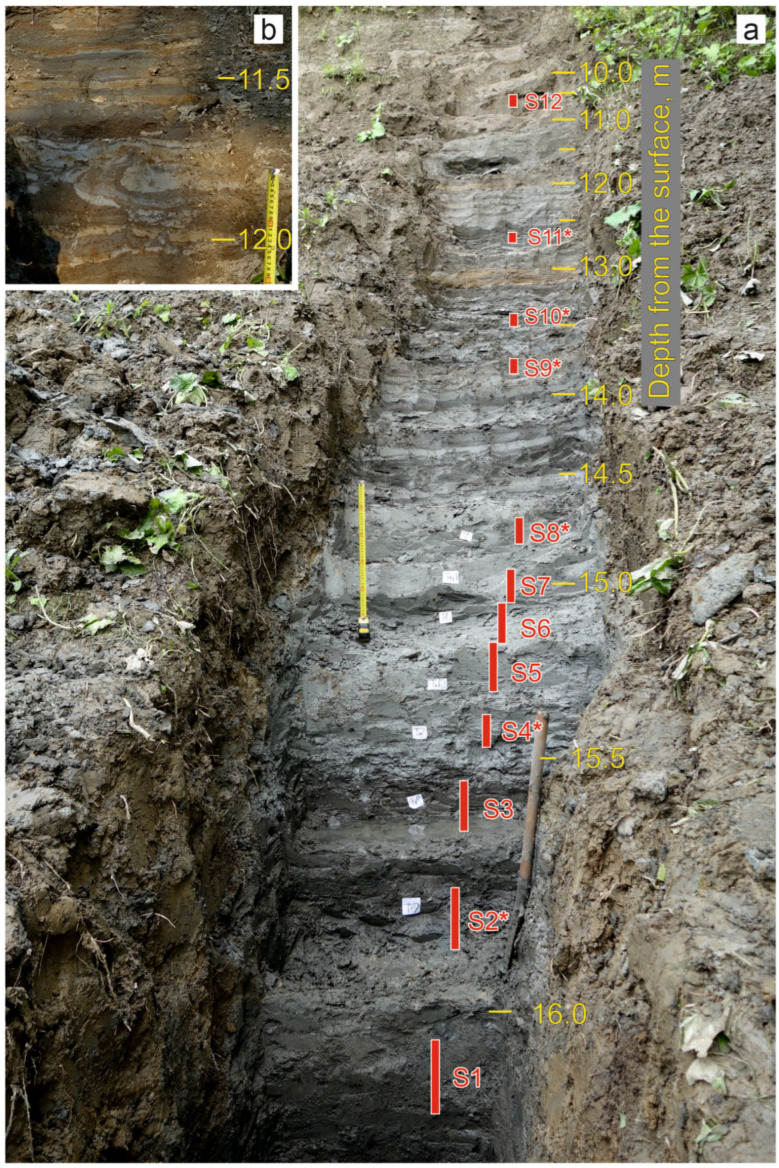
Photograph of the Kebezen section showing sampling location. Samples marked with an asterisk have a radiocarbon date. (**a**)—Main section, (**b**)—cryoturbation in the upper part of the additional section (10 m to the left from the main section).

**Figure 4 insects-16-00321-f004:**
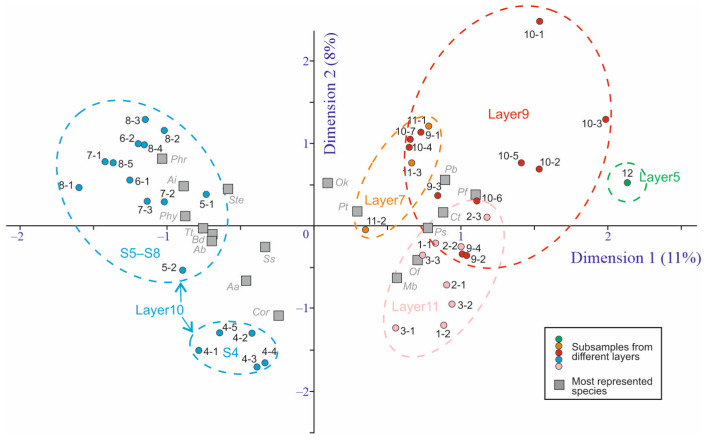
Standard correspondence analysis graph based on 64 Coleoptera species and 40 subsamples from the Kebezen site. Species represented in more than 8 subsamples are shown in the graph: ***Aa***—*Agonum alpinum*, ***Ab***—*Arpedium* cf. *brachypterum*, ***Ai***—*Augyles intermedius*, ***Bd***—*Bembidion difficile*, ***Cor***—*Coryphium* sp.1, ***Ct***—*Carphoborus teplouchovi*, ***Mb***—*Mannerheimia* cf. *brevipennis*, ***Of***—*Olophrum* cf. *fuscum*, ***Ok***—*Ochtebius kaninensis*, ***Pb***—*Pterostichus* cf. *burjaticus*, ***Pf***—*P.* cf. *fulvescens*, ***Pt***—*P. triseriatus*, ***Phr***—*Phratora* sp., ***Phy***—*Phyllodrepa* sp.1, ***Ps***—*Polygraphus subopacus*, ***Ss***—*Simplocaria semistriata*, ***Ste***—*Stenus* sp.1, ***Tt***—*Trechus* cf. *toroticus*.

**Figure 5 insects-16-00321-f005:**
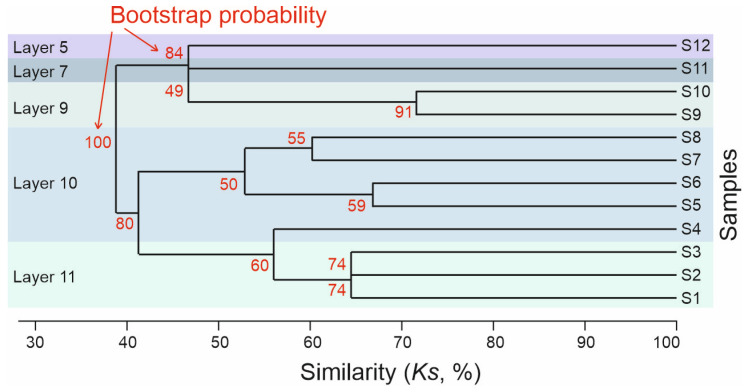
Similarity dendrogram of Coleoptera assemblages from the Kebezen site, calculated according to the stratigraphic position of the samples. UPGMA method, Simpson’s index.

**Figure 6 insects-16-00321-f006:**
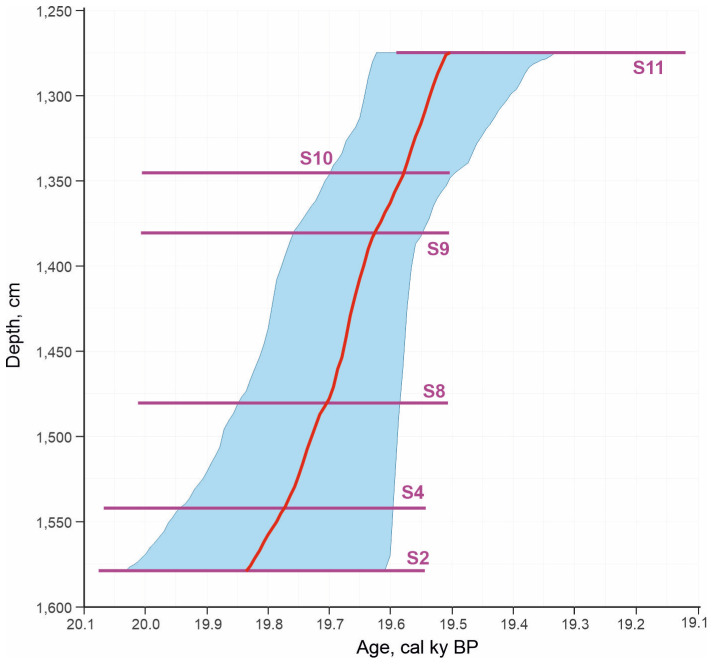
Age–depth model of deposition of the Kebezen section.

**Figure 7 insects-16-00321-f007:**
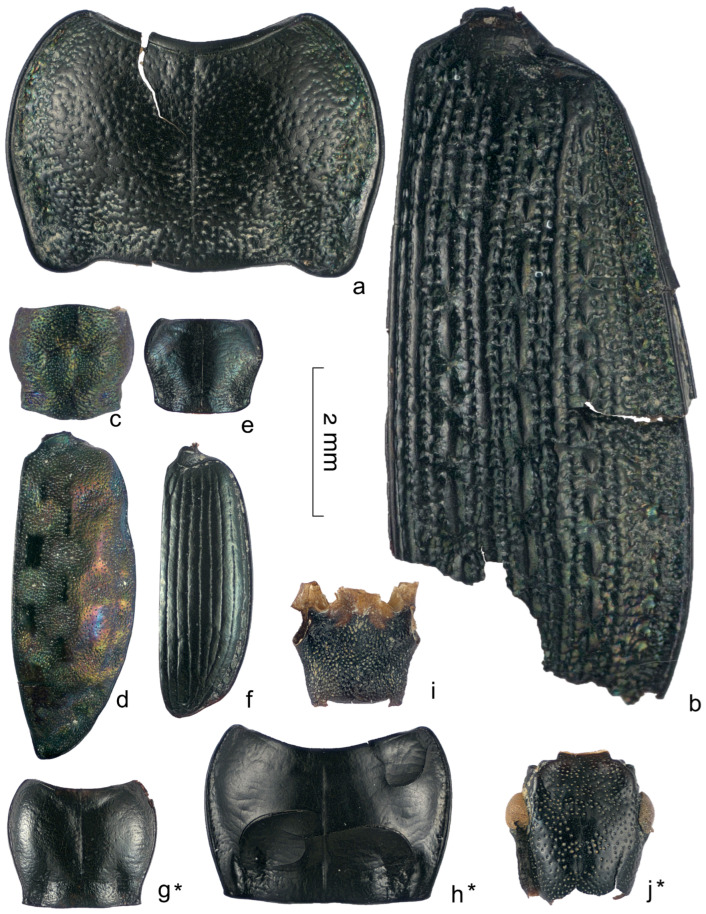
Carabidae (**a**–**h**), Cantharidae (**i**), and Pythidae (**j**) fragments from the Kebezen site. (**a**,**b**)—*Carabus arvensis* (S8), (**c**,**d**)—*Elaphrus angusticollis* (S8), (**e**,**f**)—*Agonum alpinum* (S2, S4), (**g**)—*Pterostichus* cf. *burjaticus* (S2), (**h**)—*P. triseriatus* (S6), (**i**)—*Podabrus alpinus* (S5), (**j**)—*Pytho depressus* (S5). * Species recorded in Pleistocene of western Siberia for the first time.

**Figure 8 insects-16-00321-f008:**
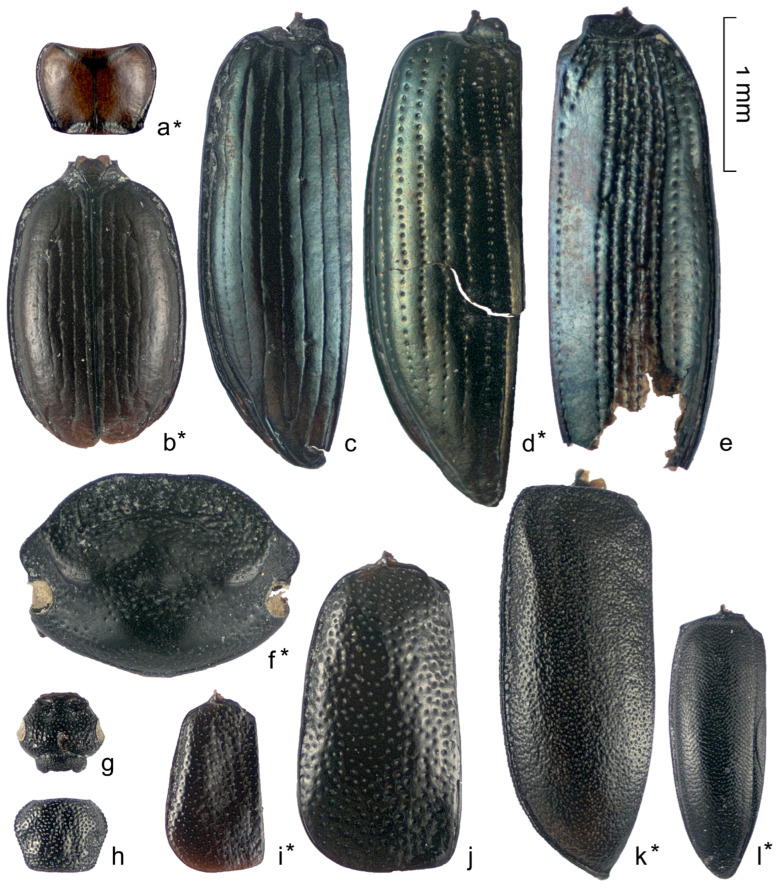
Carabidae (**a**–**e**), Scarabaeidae (**f**), Staphylinidae (**g**–**j**), Heteroceridae (**k**), and Leiodidae (**l**) fragments from the Kebezen site. (**a**,**b**)—*Trechus* cf. *toroticus* (S8), (**c**)—*Bembidion difficile* (S5), (**d**)—*B. bipunctatum* (S10), (**e**)—*Notiophilus fasciatus* (S2), (**f**)—*Aphodius ater* (S1), (**g**,**h**)—*Coryphium* sp.1 (S4), (**i**)—*Arpedium* cf. *brachypterum* (S4), (**j**)—*Olophrum* cf. *fuscum* (S1), (**k**)—*Augyles intermedius* (S7), (**l**)—*Colon* cf. *bidentatum* (S8). * Species recorded in Pleistocene of western Siberia for the first time.

**Figure 9 insects-16-00321-f009:**
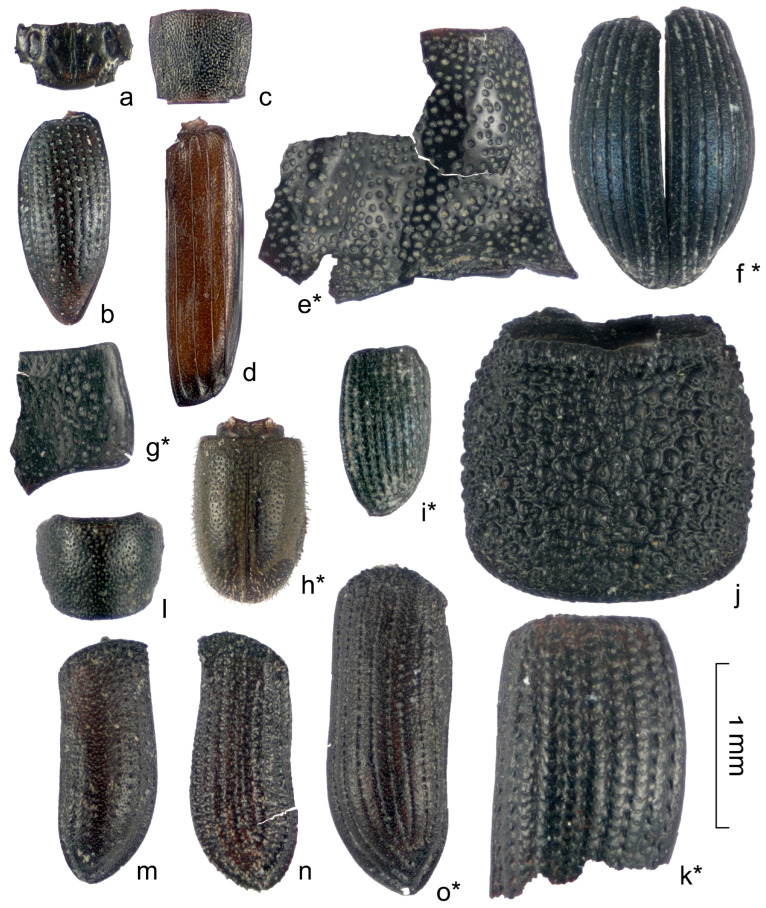
Hydraenidae (**a**,**b**), Laemophloeidae (**c**,**d**), Elateridae (**e**), Brentidae (**f**), Chrysomelidae (**g**), Ciidae (**h**), Curculionidae (**i**–**k**), and Scolytidae (**l**–**o**) fragments from the Kebezen site. (**a**,**b**)—*Ochtebius kaninensis* (S10, S4), (**c**,**d**)—*Leptophloeus alternans* (S9, S2), (**e**)—*Denticollis acuticollis* (S1), (**f**)—*Holotrichapion aethiops* (S10), (**g**)—*Sternoplatys* cf. *fulvipes* (S10), (**h**)—*Sulcacis nitidus* (S2), (**i**)—*Ceutorhyncus cochleariae* (S10), (**j**)—*Otiorhynchus grandineus* (S7), (**k**)—*Notaris acridulus* (S2), (**l**,**m**)—*Polygraphus subopacus* (S1, S2), (**n**)—*Carphoborus teplouchovi* (S10), (**o**)—*Xylechinus pilosus* (S9). * Species recorded in Pleistocene of western Siberia for the first time.

**Figure 10 insects-16-00321-f010:**
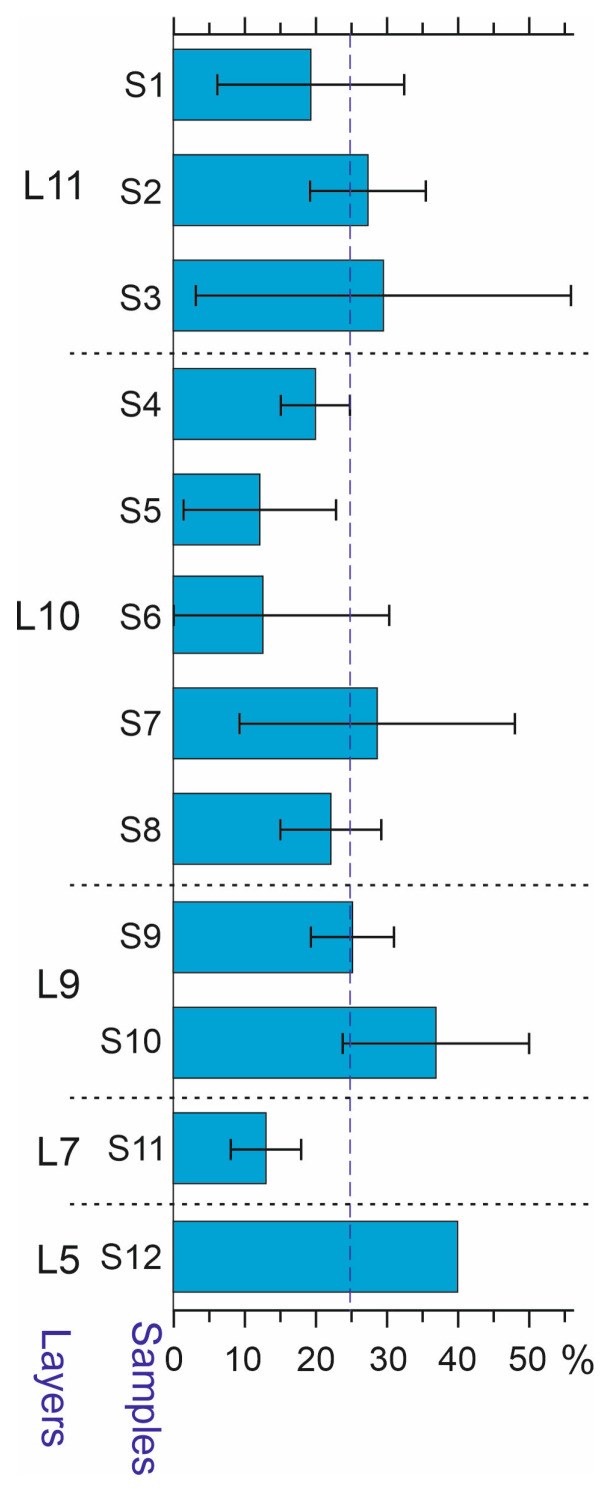
High-elevation species of Coleoptera in the samples of the Kebezen site, % (mean and standard deviation). The horizontal dotted line shows the average level. L5–L11—Layers 5–11.

**Figure 11 insects-16-00321-f011:**
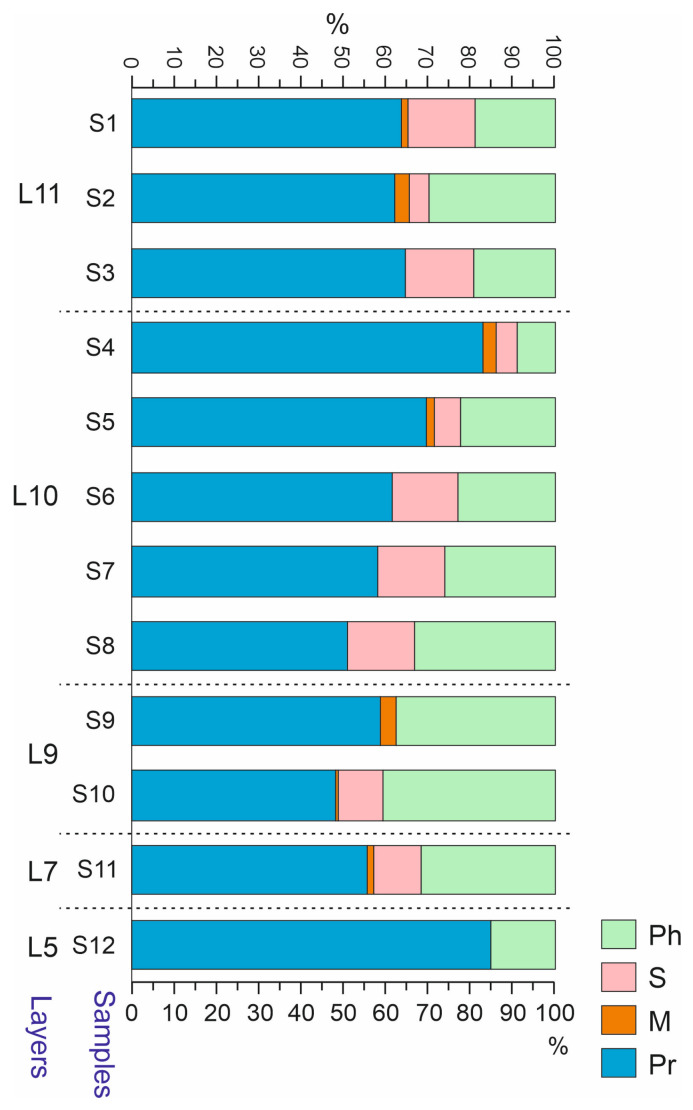
Trophic composition of Coleoptera in the samples of the Kebezen site, %. Ph—phytofagous, S—sapro- and coprophagous, M—mycetophagous, Pr—predatory. L5–L11—Layers 5–11.

**Figure 12 insects-16-00321-f012:**
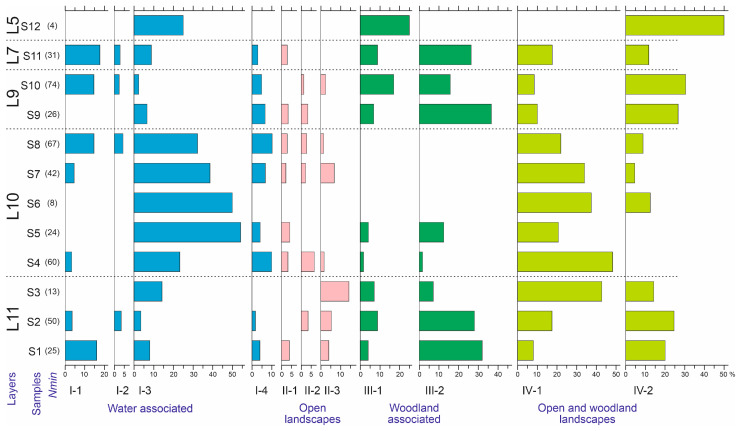
Abundance of Coleoptera ecological groups in the samples of the Kebezen site, %. L5–L11—Layers 5–11. I-1–IV-2— Ecological groups, for a description see Table 5.

**Figure 13 insects-16-00321-f013:**
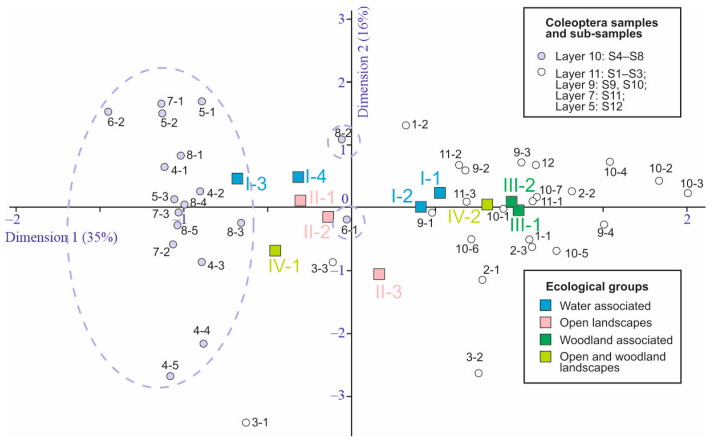
Standard correspondence analysis graph based on Coleoptera ecological groups in the subsamples of the Kebezen site. I-1–IV-2— Ecological groups, for a description see Table 5.

**Table 1 insects-16-00321-t001:** Description of the Kebezen section.

Layer No	Depth of Bed, m	Elevation, m	Thickness, m	Description
1	0.0–0.2	16.95–17.15	0.2	Modern soil.
2	0.2–0.8	16.35–16.95	0.6	Light grey sandy loam, dense.
3	0.8–6.8	10.35–16.35	6.0	Brown loam, in some places of a red-brown color.
4	6.8–10.4	6.75–10.35	3.6	Interlayering of light grey clays and reddish grey clays with spots and interlayers of ferrugination.
5	10.4–11.8	5.35–6.75	1.4	Interlayering of dark grey clays with lenses of plant detritus and light grey clays with spots of ferrugination (*sample S12*).
6	11.8–12.2	4.95–5.35	0.4	Dark grey loam with a layer (0.05 m) of dense rufous clay in the middle.
7	12.2–13.0	4.15–4.95	0.8	Rhythmic horizontal alternation of light-grey clays and dark-grey clays, including lenses and layers of plant detritus (*sample S11*).
8	13.0–13.2	3.95–4.15	0.2	Dense rufous sandy loam.
9	13.2–14.55	2.6–3.95	1.35	Rhythmic horizontal alternation of light-grey clays and dark-grey clays, including lenses and layers of plant detritus (*samples S9, S10*).
10	14.55–15.55	1.6–2.6	1.0	Dense light grey non-bedding clay with inclusions of plant detritus and twigs (*samples S4–S8*).
11	15.55–17.15	0.0–1.6	1.6	Dense dark grey clay with plant detritus, which goes under the water edge in the creek. In the upper part of the layer, there are interlayers of dark coarse-grained sand (*samples S1–S3*).

**Table 2 insects-16-00321-t002:** Samples of the Kebezen site for entomological and radiocarbon analysis.

Layer No	Sample No	Depth of Sample, m	Number of Subsamples	Subsample Volume, Liters	Radiocarbon Date, ky BP; (Laboratory Code)	Calibrated Date, cal ky BP
5	S12	10.6–10.8	1	5	–	–
7	S11	12.7–12.8	3	7	16.070 ± 0.1 (SPb-4072)	19.120–19.590
9	S10	13.4–13.5	7	1–5	16.353 ± 0.1 (SPb-3761)	19.503–20.006
	S9	13.75–13.85	4	1	16.355 ± 0.1 (SPb-4073)	19.505–20.008
10	S8	14.75–14.85	5	10	16.357 ± 0.1 (SPb-4074)	19.507–20.011
	S7	14.95–15.05	3	10	–	–
	S6	15.05–15.15	2	10	–	–
	S5	15.15–15.25	3	10	–	–
	S4	15.4–15.45	5	5	16.400 ± 0.1 (SPb-4075)	19.542–20.069
11	S3	15.55–15.65	3	10	–	–
	S2	15.75–15.85	3	10	16.407 ± 0.1 (SPb-4076)	19.546–20.078
	S1	16.05–16.15	2	10	–	–

**Table 3 insects-16-00321-t003:** Representation of subfossil invertebrates in the Kebezen site.

Taxa		In Total	S1	S2	S3	S4	S5	S6	S7	S8	S9	S10	S11	S12
Gastropoda		+	+	–	–	+	–	–	+	+	+	+	+	–
Branchiopoda:	Cladocera	–	–	–	–	–	–	–	–	–	–	–	–	–
	Ostracoda	–	–	–	–	–	–	–	–	–	–	–	–	–
Arachnida:	Oribatida	+	+	–	–	–	+	–	+	–	–	–	+	–
	Aranei	26	0	0	0	0	4	1	11	6	1	1	2	0
Insecta:	Hemiptera	10	0	0	0	1	0	0	5	4	0	0	0	0
	Coleoptera	1405	108	136	48	232	85	23	141	251	78	185	104	18
	Hymenoptera	41	2	0	0	5	4	0	5	2	4	12	6	1
	Diptera	72	3	3	9	18	7	0	13	1	1	5	12	0

(+)—Found in the sample but not counted, (–)—not found in the sample.

**Table 4 insects-16-00321-t004:** Indexes of biodiversity of subfossil Coleoptera from the Kebezen site.

Index	In Total	S1	S2	S3	S4	S5	S6	S7	S8	S9	S10	S11	S12
*Nmin*	779	61	76	30	130	45	13	70	125	45	113	58	13
*Nmin*/*V*, ind./liter	5.6 ± 0.9	3.1	2.5 ± 1.0	1.0 ± 0.1	5.2 ± 2.5	1.5 ± 0.4	0.65	2.3 ± 0.6	2.5 ± 0.7	11.3 ± 7.4	11.7 ± 4.0	2.8 ± 0.7	2.6
Species number	105	31	32	17	33	23	11	26	38	22	39	34	10
Families number	21	8	13	5	13	9	6	8	11	8	11	11	4
Simpson (1–*D*)	0.974	0.923	0.967	0.950	0.948	0.918	0.964	0.939	0.960	0.944	0.963	0.976	0.944
Shannon (*H*)	4.06	3.15	3.51	2.85	3.22	2.98	2.51	3.06	3.45	3.06	3.53	3.63	2.22

**Table 5 insects-16-00321-t005:** Ecological groups of Coleoptera assemblages from the Kebezen site.

Ecology		Definition	Species
Type	Group		
I. Water-associated	1. Aquatic	Spend the majority of their adult life in water	*Hydroporus* sp., *Helophorus praenanus*, *H. sibiricus*, *Ochthebius kaninensis*, *O. flavipes*
	2. Peat bogs, fens, and marshland	Habitat across a variety of semi-aquatic environments, such as mires, marsh, swamp, and fens	*Phaedon armoraciae*, *Notaris acridulus*, *Ceutorhynchus cochleariae*
	3. Water edge	Usually live at the edge of ponds, lakes, or slow streams	*Nebria gyllenhali*, *Elaphrus angusticollis*, *E. riparius*, *Dyschiriodes nigricornis*, *Bembidion bipunctatum*, *B. difficile*, *B. sajanum*, *Augyles intermedius*
	4. Moist localities	Live in moist localities such as floodplains but not strictly ‘wetlands’	*Loricera pilicornis*, *Psammoporus matalini*, *Simplocaria semistriata*, *Hypnoidus rivularius*, *Lochmaea caprea*, *Tournotaris bimaculata*, *Dorytomus taeniatus*, *Tachyerges stigma*/*pseudostigma*
II. Open landscapes	1. Meadow	Prefer grassland and different types of meadows	*Sitona lineellus*, *Phyllobius pomaceus*
	2. Alpine meadow	High-elevation meadow species	*Trechus lomakini*, *Coccinella nivicola*
	3. Tundra	Live in tundra and nival belt of mountains	*Diacheila polita*, *Bembidion dauricum*, *Hemitrichapion tschernovi*
III. Woodland-associated	1. Woodland	Live in woodland	*Notiophilus fasciatus*, *Carabus aeruginosus*, *C. obovatus*, *P. maurusiacus*, *P. monticoloides*, *Podabrus alpinus*/*annulatus*, *Denticollis acuticollis*, *Sulcacis nitidus*, *Sternoplatys fulvipes*, *Holotrichapion aethiops*
	2. Xylophagous and xylobionts	Develop in trees	*Megatoma undata*, *Leptophloeus alternans*, *Pytho depressus*, *Pissodes gyllenhali*, *Phloeotribus spinulosus*, *Carphoborus teplouchovi*, *Polygraphus polygraphus*, *P. subopacus*, *Xylechinus pilosus*
IV. Open and woodland landscapes	1. Boreo-alpine	Live in boreal forests and alpine meadows	*Trechus toroticus*, *Pterostichus fulvescens*, *P. triseriatus*, *P. ehnbergi*, *Agonum alpinum*, *Coryphium* sp., *Rhagonycha nigriventris*, *Eutrichapion rhomboidale*, *Notaris* sp.1, *Trichalophus maklini*, *Otiorhynchus grandineus*
	2. Forest–tundra	Live in boreal forests and tundra	*Carabus arvensis*, *Pterostichus brevicornis*, *P. burjaticus*, *P. subaeneus*, *Byrrhus mordkovitshi*, *Boreohypera diversipunctata*
	3. Eurytopic	Live in different biotopes or preferences are unclear	*Arpedium brachypterum*, *Mannerheimia brevipennis*, *Olophrum fuscum*, *Aphodius ater*

## Data Availability

Material was deposited in the collection of the Institute of Systematics and Ecology of Animals.

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
