# Peer review of "Subfossil Insects of the Kebezen Site (Altai Mountains): New Data on the Last Deglaciation Environment"

_insects, 2025, doi:10.3390/insects16030321_

Round 1
Reviewer 1 Report
Comments and Suggestions for Authors
All my comments are in the edited manuscript.

The English is in need of revision. I have included all suggestions for language improvements in the edited text.
Author Response
We sincerely thank the reviewer for the comments and improvement of the English language. We have corrected the manuscript according to the reviewer's comments.
Reviewer 2 Report
Comments and Suggestions for Authors
Simple Summary: It briefly describes the main points – the study’s aim, the location, the age of the sediments, the basic findings about the entomofauna, and the principal conclusions about climate and vegetation. However, one might consider adding a more explicit statement in the Simple Summary about why this study is important, for instance for paleoecology or for understanding postglacial processes in the region.
Lines 177–178: It might sound better to say, “The lake sediment outcrop on the left bank… was first described by Baryshnikov in 1981…”.
Line 187: The phrase “surveyed by us…” is somewhat informal. A possible rephrasing is, “We surveyed the Kebezen site on the Turachak stream in July 2019…”.
Line 253: Merely stating that the authors used “hierarchical clustering with the Simpson similarity coefficient” is insufficient for full transparency. There is no information about the linkage method (e.g., UPGMA, as hinted later). I also miss a rationale for choosing the Simpson coefficient. While it is reasonable in this context, given that the shared absence is less influential, the reader might at least expect a brief note on why the authors did not select, for example, the Bray–Curtis index more commonly used in general biology. It would also be appropriate to mention whether the authors performed any validation (e.g., bootstrap) within the clustering – It appears later in the results but should be stated in the methods.
Line 255: Regarding correspondence analysis, it is not specified how the authors dealt with species of very low abundance (rare taxa), nor whether they performed any data transformations. Moreover, some sort of permutation tests to verify the significance of CA axes or to test hypotheses, while not strictly mandatory, are commonly used when interpreting results. Readers might even appreciate a note that the authors did not test for axis significance and only used a visual interpretation of the ordination diagram.
Lines 268–269: Regardless of the detailed procedure, the authors should state what type of model they used – whether default settings or otherwise, priors specification, whether they used the IntCal20 curve for calibration, and how many iterations (or what Markov chain settings) were applied… Even the phrase “an age-depth model was constructed” is very general. A reviewer would expect a methodological note as to whether, for example, the authors generated mean or median depth estimates, and how they interpreted the resulting curve in the text (e.g., polynomial curve, linear interpolation, Bayesian random walk, etc.).
Line 281: There may have been an unintentional minor discrepancy: here the authors write “no fewer than 104 species,” whereas in the abstract (and elsewhere) they mention 105. Possibly one indeterminate taxon (sp./cf.) was counted in the abstract, which the authors then left out in the main text? I recommend unifying the numbers in the text and, if necessary, briefly explaining why there is a difference.
Line 284: Previously, “Scolytidae” used to be cited as a separate family. According to modern classification, bark beetles (traditionally Scolytidae) are placed as the subfamily Scolytinae within the family Curculionidae (i.e., weevils).
Figure 4: Consider adding a few words of interpretation to the caption: “Points represent sub-samples. The first two axes explain XX% of the variation...”. This way, readers can better gauge how representative the two-dimensional projection is.
Figure 5: UPGMA vs. UPGA – do the authors actually mean UPGMA? Furthermore, UPGMA is a relatively older method sometimes criticized in ecological data because it can lead to certain distortions in the dendrogram (favoring clusters with higher richness, etc.). In the context of this study, it is likely a legitimate choice, but the authors might briefly justify it – obviously in the methods section.
Section 5.3 Ecological analysis: This section mixes new descriptions and figures (e.g., specific percentages in individual samples, references to new tables and graphs) with interpretations. Some of these details conflict with the usual IMRaD approach, where detailed results belong in the “Results” section, and the method for how these results were generated belongs in the “Methods,” while only interpretations should appear in the discussion. I strongly recommend revising this. For instance, dividing species into trophic groups is itself a methodological step; the numerical distribution (proportions of each group) is a result; the statistical data (specific percentages in individual samples, a graph) are effectively “results.” Additional correspondence analysis in lines 497–498 is not at all described in the methods, and its results plus graph should again appear in the results, etc.
Figure 9: The y-axis label should be aligned horizontally in consistency with the other graphs.
Lines 454–466: Here the authors forgot to italicize the Latin names.
Figure 11: It might be better to refer to proportion rather than abundance if it is expressed in percentages.
Section 5.4 Conditions…: From my perspective, the same issue persists as in the previous section – interpretation is mixed with relatively new results. For example, we encounter sedimentation rate values stated right here (1 cm/year according to the model). We also see the time context of two radiocarbon dates from previous studies (e.g., 16120 ± 80 BP, etc.) or specific information on community composition in each layer. Some of these pieces of information could have been presented earlier (in “Results”).
Author Response
Dear Reviewer, thank you for your careful reading of the manuscript and valuable comments! We have taken most of them into account and enlarged the manuscript. Explanations of our point of view on the rest are listed below.
Simple Summary: It briefly describes the main points – the study’s aim, the location, the age of the sediments, the basic findings about the entomofauna, and the principal conclusions about climate and vegetation. However, one might consider adding a more explicit statement in the Simple Summary about why this study is important, for instance for paleoecology or for understanding postglacial processes in the region.
Done
Lines 177–178: It might sound better to say, “The lake sediment outcrop on the left bank… was first described by Baryshnikov in 1981…”.
Done
Line 187: The phrase “surveyed by us…” is somewhat informal. A possible rephrasing is, “We surveyed the Kebezen site on the Turachak stream in July 2019…”.
Done
Line 253: Merely stating that the authors used “hierarchical clustering with the Simpson similarity coefficient” is insufficient for full transparency. There is no information about the linkage method (e.g., UPGMA, as hinted later). I also miss a rationale for choosing the Simpson coefficient. While it is reasonable in this context, given that the shared absence is less influential, the reader might at least expect a brief note on why the authors did not select, for example, the Bray–Curtis index more commonly used in general biology. It would also be appropriate to mention whether the authors performed any validation (e.g., bootstrap) within the clustering – It appears later in the results but should be stated in the methods.
Done. Methods was enlarged
Line 255: Regarding correspondence analysis, it is not specified how the authors dealt with species of very low abundance (rare taxa), nor whether they performed any data transformations. Moreover, some sort of permutation tests to verify the significance of CA axes or to test hypotheses, while not strictly mandatory, are commonly used when interpreting results. Readers might even appreciate a note that the authors did not test for axis significance and only used a visual interpretation of the ordination diagram.
Done. Methods was enlarged
Lines 268–269: Regardless of the detailed procedure, the authors should state what type of model they used – whether default settings or otherwise, priors specification, whether they used the IntCal20 curve for calibration, and how many iterations (or what Markov chain settings) were applied… Even the phrase “an age-depth model was constructed” is very general. A reviewer would expect a methodological note as to whether, for example, the authors generated mean or median depth estimates, and how they interpreted the resulting curve in the text (e.g., polynomial curve, linear interpolation, Bayesian random walk, etc.).
Done. Methods was enlarged
Line 281: There may have been an unintentional minor discrepancy: here the authors write “no fewer than 104 species,” whereas in the abstract (and elsewhere) they mention 105. Possibly one indeterminate taxon (sp./cf.) was counted in the abstract, which the authors then left out in the main text? I recommend unifying the numbers in the text and, if necessary, briefly explaining why there is a difference.
Inaccuracy changed. 105 is correct.
Line 284: Previously, “Scolytidae” used to be cited as a separate family. According to modern classification, bark beetles (traditionally Scolytidae) are placed as the subfamily Scolytinae within the family Curculionidae (i.e., weevils).
We adhere to the point of view of the independence of the bark beetles as a family following Morimoto & Kojima (2004) and Legalov (2024). The presence of significant morphological differences in the Scolytidae, as well as their appearance in the fossil record earlier than the primitive Curculionidae, does not allow us to place them in a single family.
Legalov A.A. (2024) Fossil history of bark-beetles (Coleoptera: Scolytidae) with descriptions of two new species, Historical Biology, 36:2, 378-388, DOI: 10.1080/08912963.2022.2157275
Morimoto K, Kojima H. 2004. Systematic position of the tribe Phylloplatypodini, with remarks on the definitions of the families Scolytidae, Platypodidae, Dryophthoridae and Curculionidae (Coleoptera: Curculionoidea). Esakia. 44:153–168. doi:10.5109/2691.
Figure 4: Consider adding a few words of interpretation to the caption: “Points represent sub-samples. The first two axes explain XX% of the variation...”. This way, readers can better gauge how representative the two-dimensional projection is.
Done. The Figure 4 was slightly changed. It was recomputed without singletons species. The most represented species and percenters of axes variation were shown in the plot.
Figure 5: UPGMA vs. UPGA – do the authors actually mean UPGMA? Furthermore, UPGMA is a relatively older method sometimes criticized in ecological data because it can lead to certain distortions in the dendrogram (favoring clusters with higher richness, etc.). In the context of this study, it is likely a legitimate choice, but the authors might briefly justify it – obviously in the methods section.
Done
Figure 9: The y-axis label should be aligned horizontally in consistency with the other graphs.
Done. The samples in Figures 9 and 10 (10 and 11 after added of Figure 6) are now arranged vertically, according to the stratigraphy.
Lines 454–466: Here the authors forgot to italicize the Latin names.
Done.
Figure 11: It might be better to refer to proportion rather than abundance if it is expressed in percentages.
Yes, but proportion (percentage) can be calculated both from abundance and from the number of species. In this case, precisely from abundance.
Section 5.3 Ecological analysis: This section mixes new descriptions and figures (e.g., specific percentages in individual samples, references to new tables and graphs) with interpretations. Some of these details conflict with the usual IMRaD approach, where detailed results belong in the “Results” section, and the method for how these results were generated belongs in the “Methods,” while only interpretations should appear in the discussion. I strongly recommend revising this. For instance, dividing species into trophic groups is itself a methodological step; the numerical distribution (proportions of each group) is a result; the statistical data (specific percentages in individual samples, a graph) are effectively “results.” Additional correspondence analysis in lines 497–498 is not at all described in the methods, and its results plus graph should again appear in the results, etc.
Section 5.4 Conditions…: From my perspective, the same issue persists as in the previous section – interpretation is mixed with relatively new results. For example, we encounter sedimentation rate values stated right here (1 cm/year according to the model). We also see the time context of two radiocarbon dates from previous studies (e.g., 16120 ± 80 BP, etc.) or specific information on community composition in each layer. Some of these pieces of information could have been presented earlier (in “Results”).
Done only partially. Improved description of the correspondence analysis in the Methods. The age-depth model figure and its description have been moved to Results. However, we left the ecological analysis in the discussion section. The radiocarbon dates given in section 5.4 are taken from the literature for sections similar to Kebezen – i.e. this is a discussion. The “pieces of specific information” presented in subchapters 5.4.1–5.4.3 are a repetition of the results already presented, on which the justification for the primary succession hypothesis is based. There are no new results here.
We agree that sections 5.3 and 5.4 do not fully correspond to the Introduction–Methods–Results–Discussion scheme, which is optimal for experimental work. However, many works on natural history, for example, taxonomic or faunistic (especially from poorly studied regions), etc., do not fit well into this scheme. Unfortunately, most journals now prohibit combining results and discussions in one section. In our opinion, the main (primary) result of work on Quaternary entomology is a list of insects with distribution according to stratigraphy. Further analysis directly depends on the primary data obtained, i.e. it is substantiated during the discussion. Thus, in this manuscript, the cold conditions are most convincingly substantiated in the Section 5.3.1 “Altitudinal distribution”. The choice of approach is due to the fact that the overwhelming majority of species from the Kebezen site live in north-eastern Altai today. Otherwise (as, for example, in the majority of late Pleistocene faunas of Southern Siberia, which have no modern analogues), such an analysis would not make sense, and instead, a geographical (zonal) distribution or the mutual climatic range method would be appropriate. Our proposed division of species into ecological groups and reconstructions of Pleistocene biotopes are also based, to a large extent, on the preferences of species in north-eastern Altai, although some of them have greater ecological valence in other regions. This would be questionable for non-analogue entomofaunas. Section 5.4 aims to explain the resulting contradiction between the high rate of sediment accumulation and the apparent dynamics in the ecological composition of beetles.
The approach used in our manuscript preserves the cause-and-effect relationships in the analysis of insect assemblages and therefore “additional results” appear during the discussion. Of course, this can be avoided by putting forward our interpretations as primary hypotheses. Perhaps such a presentation would be easier for many readers to understand. However, putting forward any specific hypotheses for a very poorly studied region is rather questionable. Moreover, warm and dry conditions (rather than cold and humid) are more likely to be expected for the initial stage of deglaciation, and non-analogues faunas (rather than similar to the modern fauna of Altai) are characteristic of the end of the Pleistocene in southern Siberia; the assumption of primary succession appears as an explanation for the regular change in ecological composition at high sedimentation rates, and not vice versa. Therefore, we invite the inquisitive reader to come to the received interpretations step by step, with a consistent immersion in the analysis.
Reviewer 3 Report
Comments and Suggestions for Authors
Dear Authors,
Your study examining subfossil insect assemblages from the Kebezen site in the Altai Mountains presents an analysis of Coleoptera fauna during the last deglaciation. The research documents 105 beetle species, including endemic taxa and species not previously recorded in Pleistocene deposits of Western Siberia. The dataset provides evidence of rapid ecological changes during the studied period, with proposed connections to megaflood events. The methodological approach combining radiocarbon dating with ecological succession models supports the chronological framework. This work adds to existing data regarding potential refugium zones in the Altai Mountains during the late Pleistocene and demonstrates the application of insect remains in paleoenvironmental studies.
My comments are mainly about the ecology, please consider.
1. Community Composition and Paleoenvironmental Reconstruction
The insect assemblage (dominated by Carabidae and Staphylinidae) from the Kebezen site shows significant differences from the Late Pleistocene insect fauna of the West Siberian Plain, yet exhibits similarities with the neighboring Lebed site and modern mid-altitude insect communities in the Altai Mountains. Does this geographical distribution pattern reflect unique microclimates and habitat heterogeneity in the Altai Mountains during the initial phase of the Last Glacial-Interglacial transition? Specifically, which environmental factors (e.g., temperature, humidity, vegetation types) have driven the regional differentiation of these insect communities?
2. Ecological Succession and Disturbance Events
The study suggests that changes in beetle ecological composition may correspond to primary succession processes triggered by megafloods. How can we quantitatively reconstruct post-flood ecosystem recovery dynamics through proportional variations of meadow/riparian species, shrub species, moss-dwelling species, and forest-associated species across different sedimentary layers? Does this succession pattern align with established recovery models for terrestrial ecosystems following flood disturbances?
3. Climate-Vegetation-Insect Coevolution
Insect data indicate the coexistence of a Picea-dominated boreal forest and alpine meadow ecosystem in this region during the early Last Glacial-Interglacial transition. What mechanisms maintained the stability of this vegetation assemblage under cold-humid climatic conditions? Does the emergence of Picea as a dominant tree species demonstrate co-evolutionary relationships with specific beetle taxa (e.g., Scolytidae)? Furthermore, does the structural similarity between this ancient ecosystem and modern mid-altitude insect communities?
As not a native speaker, I have no comment about this.
Author Response
Dear Reviewer,
Thank you for your feedback and questions, which allowed us to rethink our research and make some additions to the manuscript.
- Community Composition and Paleoenvironmental Reconstruction
The insect assemblage (dominated by Carabidae and Staphylinidae) from the Kebezen site shows significant differences from the Late Pleistocene insect fauna of the West Siberian Plain, yet exhibits similarities with the neighboring Lebed site and modern mid-altitude insect communities in the Altai Mountains. Does this geographical distribution pattern reflect unique microclimates and habitat heterogeneity in the Altai Mountains during the initial phase of the Last Glacial-Interglacial transition? Specifically, which environmental factors (e.g., temperature, humidity, vegetation types) have driven the regional differentiation of these insect communities?
Yes, indeed, our research confirms the differentiation of mountain and plain ecosystems in the late Pleistocene, caused by climatic and landscape-biotopic differences in the regions. First of all, the differences are associated with the significantly greater humidity of the northeastern Altai, against the background of the general aridity of Siberia. Necessary additions have been made to section 5.3.3 and to the Conclusion.
- Ecological Succession and Disturbance Events
The study suggests that changes in beetle ecological composition may correspond to primary succession processes triggered by megafloods. How can we quantitatively reconstruct post-flood ecosystem recovery dynamics through proportional variations of meadow/riparian species, shrub species, moss-dwelling species, and forest-associated species across different sedimentary layers? Does this succession pattern align with established recovery models for terrestrial ecosystems following flood disturbances?
In general, the scheme we presented corresponds to known models of restoration of forest ecosystems after significant (catastrophic) impacts on the ecosystem. Megafloods (carrying a huge amount of sediment in addition to water) are better compared to mudflows or avalanches than to floods. Unfortunately, we were unable to find in the literature examples of catastrophic floods in taiga forests, followed by a long period without floods, allowing us to trace the course of succession.
- Climate-Vegetation-Insect Coevolution
Insect data indicate the coexistence of a Picea-dominated boreal forest and alpine meadow ecosystem in this region during the early Last Glacial-Interglacial transition. What mechanisms maintained the stability of this vegetation assemblage under cold-humid climatic conditions? Does the emergence of Picea as a dominant tree species demonstrate co-evolutionary relationships with specific beetle taxa (e.g., Scolytidae)? Furthermore, does the structural similarity between this ancient ecosystem and modern mid-altitude insect communities?
The question posed is certainly very interesting. However, it goes beyond the scope of our study, since its solution requires an analysis of much larger material both in time and geographic scales.